EMBO
Molecular Medicine

# D-Aspartate treatment attenuates myelin damage and stimulates myelin repair

Valeria de Rosa[1] (iD), Agnese Secondo[1], Anna Pannaccione[1], Roselia Ciccone[1], Luigi Formisano[1], Natascia Guida[2], Roberta Crispino[3], Annalisa Fico[4], Roman Polishchuk[3], Antimo D'Aniello[1], Lucio Annunziato[2] & Francesca Boscia[1],* (iD)

## Abstract

Glutamate signaling may orchestrate oligodendrocyte precursor cell (OPC) development and myelin regeneration through the activation of glutamate receptors at OPC-neuron synapses. D-Aspartate is a D-amino acid exerting modulatory actions at glutamatergic synapses. Chronic administration of D-Aspartate has been proposed as therapeutic treatment in diseases related to myelin dysfunction and NMDA receptors hypofunction, including schizophrenia and cognitive deficits. Here, we show, by using an *in vivo* remyelination model, that administration of D-Aspartate during remyelination improved motor coordination, accelerated myelin recovery, and significantly increased the number of small-diameter myelinated axons. Chronically administered during demyelination, D-Aspartate also attenuated myelin loss and inflammation. Interestingly, D-Aspartate exposure stimulated OPC maturation and accelerated developmental myelination in organotypic cerebellar slices. D-Aspartate promoting effects on OPC maturation involved the activation of glutamate transporters, AMPA and NMDA receptors, and the Na$^+$/Ca$^{2+}$ exchanger NCX3. While blocking NMDA or NCX3 significantly prevented D-Aspartate-induced [Ca$^{2+}$]$_i$ oscillations, blocking AMPA and glutamate transporters prevented both the initial and oscillatory [Ca$^{2+}$]$_i$ response as well as D-Aspartate-induced inward currents in OPC. Our findings reveal that D-Aspartate treatment may represent a novel strategy for promoting myelin recovery.

**Keywords** cuprizone; D-Aspartate; NCX3; oligodendrocytes; remyelination
**Subject Categories** Neuroscience; Pharmacology & Drug Discovery; Stem Cells

## Introduction

A significant interest in new multiple sclerosis (MS) therapeutics is the identification of novel pharmacological compounds able to stimulate the remyelination process by boosting oligodendrocyte precursors cells (OPC) to form new myelin before axons become irreversibly damaged (Franklin & Ffrench-Constant, 2017). Furthermore, remyelination therapy is emerging as promising strategy not only for restoration of axonal conduction properties that are lost following demyelination, but also to afford neuroprotection thereby preventing disease progression and clinical disability associated with MS or other demyelinating diseases (Mei *et al*, 2016).

There is now clear evidence that neuronal activity may regulate OPC differentiation and myelination and is essential for instructing myelin regeneration following demyelination (Gibson *et al*, 2014; Gautier *et al*, 2015; Almeida & Lyons, 2017). In fact, demyelinated axons, when still active, by releasing glutamate at OPC-neuron synapse may instruct OPC to differentiate into new myelinating oligodendrocytes (Bergles *et al*, 2000; Lundgaard *et al*, 2013; Micu *et al*, 2017). In this regard, glutamate signaling through α-amino-3-hydroxy-5-methyl-4-isoxazolepropionic acid receptor (AMPA) and N-Methyl-D-Aspartate (NMDA) receptors has garnered the main interest as inductive signal prompting OPC to differentiate and remyelinate (Yuan *et al*, 1998; Li *et al*, 2013; Fannon *et al*, 2015; Gautier *et al*, 2015). In line, mounting evidence suggests that the activation of Ca$^{2+}$-dependent pathways through glutamate receptors or the Na$^+$/Ca$^{2+}$ exchanger NCX3 may influence oligodendrocyte maturation, myelin synthesis, and remyelination processes (Boscia *et al*, 2012, 2016a; Martinez-Lozada *et al*, 2014; Casamassa *et al*, 2016; Friess *et al*, 2016).

Recently, D-amino acids are emerging as molecules with important physiological roles in neurons and glial cells (D'Aniello, 2007; Errico *et al*, 2012). Remarkably, D-Aspartate (D-Asp) has been detected in considerable levels in the central nervous and reproductive systems of vertebrates and invertebrates and plays a relevant role in the endocrine system and during nervous system development (Schell *et al*, 1997). Indeed, D-Asp regulates steroidogenesis and the release of many hormones not only in endocrine glands, but also in the brain (Ota *et al*, 2012; Di Fiore *et al*, 2018). In the rodent and human brain, free D-Asp levels are high between embryonic and early postnatal stages but then drastically decrease, concomitantly with the increased

1 Division of Pharmacology, Department of Neuroscience, Reproductive and Dentistry Sciences, School of Medicine, Federico II University of Naples, Naples, Italy
2 IRCCS SDN, Naples, Italy
3 Telethon Institute of Genetics and Medicine (TIGEM), Naples, Italy
4 Institute of Genetics and Biophysics "A. Buzzati-Traverso", Consiglio Nazionale delle Ricerche, Naples, Italy
*Corresponding author. Tel: +39 81 7463326; Fax: +39 0817463323; E-mail: boscia@unina.it

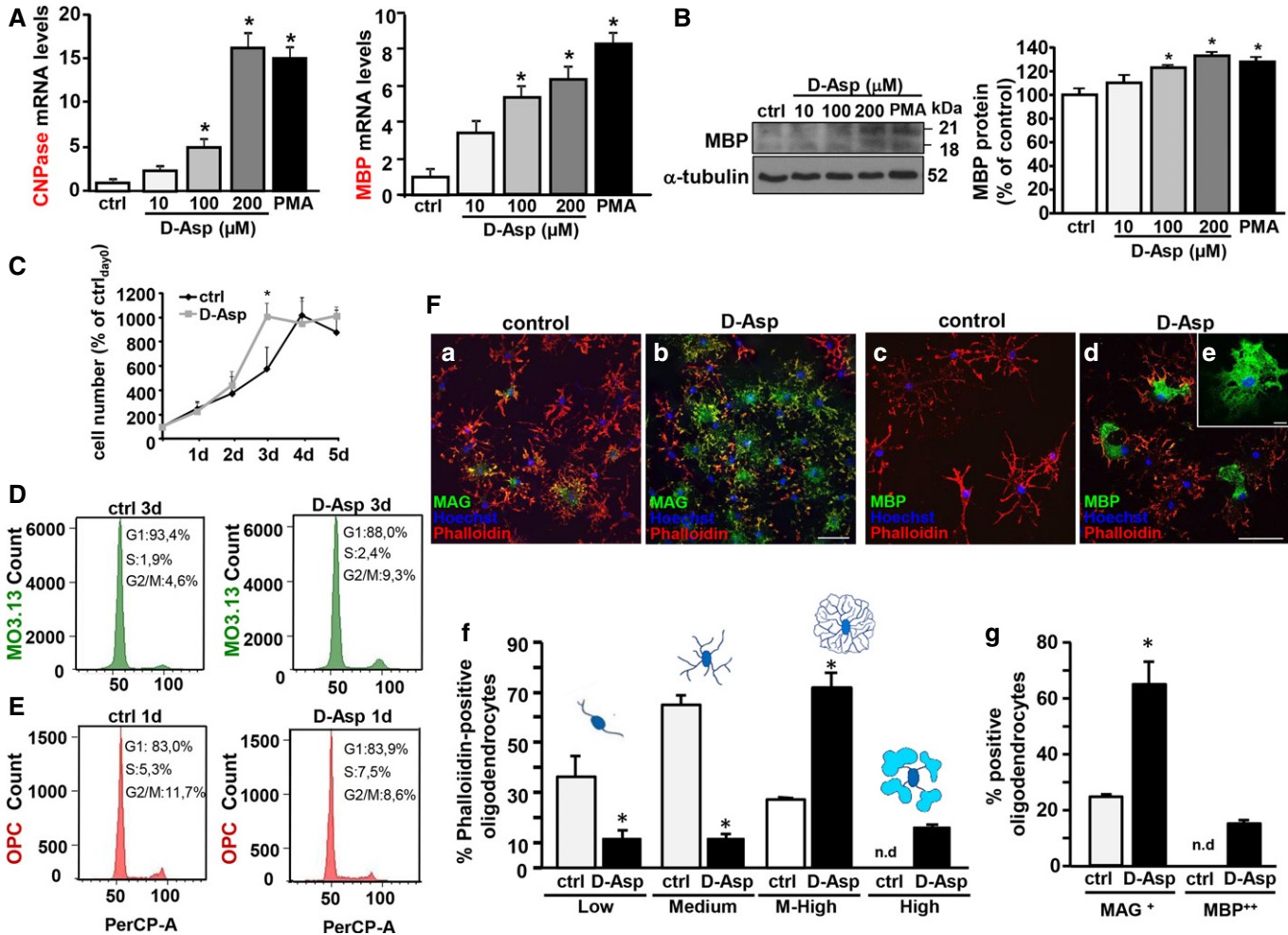

**Figure 1. Effects of D-Asp exposure on OPC proliferation and differentiation.**

A   RT–PCR of CNPase (left) and MBP (right) mRNAs expression in MO3.13 precursors under control conditions and following 10–200 μM D-Asp exposure for 3 days. Graphs show quantification of ratio of CNPase, and MBP to L19.

B   Western blotting (left) and densitometric analysis (right) of MBP expression in the absence or in the presence of 10–200 μM D-Asp exposure for 5 days.

C   Cell growth analysis of human MO3.13 oligodendrocytes in the absence or in the presence of 200 μM D-Asp for 1–5 days. The density of MO3.13 oligodendrocytes was daily recorded through trypan blue dye exclusion. Mean of daily measurements was recorded. The data of each experimental group were normalized to the density of cells plated at day 0 and expressed as percentage of $ctrl_{day0}$.

D   FACS-based cell cycle distribution analysis after PI incorporation of MO3.13 oligodendrocytes in the absence or in the presence of 200 μM D-Asp for 3 days. Representative FACS plots of biological replicates are shown ($n = 5$ independent experimental sessions).

E   FACS-based cell cycle distribution analysis after PI incorporation of rat primary OPC in the absence or in the presence of 100 μM D-Asp for 1 day. Representative FACS plots of biological replicates are shown ($n = 3$ independent experimental sessions).

F   Confocal microscopic images displaying the coexpression of MAG or MBP with phalloidin-594 (a, b and c, d, respectively) in rat primary OPC cultured in the absence or in the presence of 100 μM D-Asp for 4 days. (e) Single representative MBP++ cell. Scale bars: 50 μm in (a–d); 10 μm in (e). (f, g) Quantitative analysis of Alexa594-phalloidin-positive oligodendrocytes (f) or MAG+ and MBP++ oligodendrocytes (g) scored in control and D-Asp-exposed cells. Alexa594-phalloidin-positive oligodendrocytes were scored in four categories, according to their morphological complexity. In each category, data were normalized on the total number of oligodendrocytes.

Data information: The values represent the means ± SEM. Level of significance was determined by using: in (A) left panel, one-way ANOVA $P < 0.0001$ followed by Tukey's *post hoc* test, *$P < 0.05$ versus control ($n = 3$); (A) right panel, one-way ANOVA $P < 0.0001$ followed by Tukey's *post hoc* test, *$P < 0.05$ versus control ($n = 3$); (B) one-way ANOVA $P = 0.0001$ followed by Bonferroni *post hoc* test, *$P < 0.05$ versus control ($n = 3$); (C) for each day, two-tailed Student's *t*-test, *$P < 0.05$ versus control at 3 days ($n = 5$); (F) for each category, two-tailed Student's *t*-test, *$P < 0.05$ versus control ($n = 3$). See the exact *P*-values from comparisons tests in Appendix Table S1. Source data are available online for this figure.

expression of its degrading enzyme D-Aspartate-oxidase (DDO; D'Aniello *et al*, 1993; Errico *et al*, 2012).

Present findings support a neuromodulatory action for D-Asp at glutamatergic synapses. In fact, D-Asp may act as an endogenous agonist at NMDA receptors and may also interfere with NMDA-independent pathways (Gong *et al*, 2005; D'Aniello *et al*, 2011; Krashia *et al*, 2016; Sacchi *et al*, 2017). For instance, D-Asp is involved in NMDA receptor-dependent synaptic plasticity and learning and

memory processes (Errico *et al*, 2008a, 2011, 2014; Topo *et al*, 2010). For its steroidogenic effect on gonads and modulatory effects on memory processes, it is currently delivered as supplement for infertility or cognition, and it has been proposed as therapeutic agent for diseases related to NMDA receptor hypofunction (Errico *et al*, 2008b). The observation that D-Asp was detected in considerable levels in the white matter (Fisher *et al*, 1986) and it may influence glutamate receptor signaling and brain plasticity (Errico *et al*, 2008a) was what led us to investigate the effects of D-Asp treatment on oligodendrocytes both *in vitro*, during OPC differentiation, myelination, and remyelination, and *in vivo*, in mice fed with the copper chelator cuprizone, an *in vivo* model of myelin damage and repair.

Collectively, our results show that D-Aspartate treatment, by influencing calcium signaling via the concerted activation of glutamate transporters, AMPA and NMDA receptors, and NCX3 exchangers in oligodendrocytes, might produce beneficial effects during demyelination and remyelination.

# Results

### D-Aspartate exposure stimulates oligodendrocyte differentiation

To investigate the effect of D-Asp during oligodendrocytes differentiation, human oligodendrocyte MO3.13 precursors or rat primary OPC was exposed to D-Asp and then analyzed for myelin marker expression. RT–PCR experiments revealed that, when MO3.13 progenitors were exposed to 10–200 μM D-Asp or phorbol-12-myristate-13-acetate (PMA) for 3 days, a significant dose-dependent increase in 2′,3′-cyclic-nucleotide 3′-phosphodiesterase (CNPase) and myelin basic protein (MBP) transcripts was observed (Fig 1A). In accordance, 100–200 μM D-Asp exposure for 5 days upregulated MBP protein levels in MO3.13 oligodendrocytes, as revealed by Western blotting (Fig 1B). Analysis of cell growth in MO3.13 progenitors revealed that the density of D-Asp-treated cells on day 3 was significantly higher compared to untreated cells (Fig 1C). After 4 days, the percentage of D-Asp-treated cells, but not those of untreated, remain unaltered compared to the number of cells recorded at 3 days. At later time points, after 5 days, the number of D-Asp-treated cells, as well as those of untreated cultures, remained stable compared to the cell number recorded at 4 days (Fig 1C). In agreement with cell growth profile showing an increased proliferation of MO3.13 progenitors during D-Asp treatment, cell cycle distribution analysis by quantitative flow cytometry showed that D-Asp exposure for 3 days, but not for 1 or 2 days (data not shown), induced a G1-phase reduction before S-phase progression compared to untreated cells, which was accompanied by an accumulation in G2/M-phase (9.3% D-Asp-treated cells versus 4.6% control; Fig 1D). Interestingly, cell cycle distribution analysis on rat primary OPC exposed to D-Asp showed a significant reduction in G2/M-phase cell population if compared to untreated controls. This effect was already observed by 24 h of D-Asp exposure (Fig 1E) and persisted at 48 and 72 h (data not shown), thus suggesting that D-Asp treatment significantly reduced proliferation in rat primary OPC. Moreover, these findings also indicated that different mechanism of induction of oligodendrocyte differentiation can be observed with D-Asp exposure in clonal MO3.13 precursors and primary OPC cultures.

Confocal analysis performed with Alexa594-phalloidin and myelin-associated glycoprotein (MAG) or MBP antibodies revealed that primary OPC cultured in the presence of 100 μM D-Asp for 4 days displayed a significant lower percentage of cells with low and medium morphology compared to untreated cells (Fig 1F). By contrast, the percentage of cells displaying medium–high morphology, which were intensely stained by MAG antibodies, was significantly higher in D-Asp-treated compared to untreated cultures (Fig 1F). Finally, oligodendrocytes showing compact MBP[++]-positive regions were observed only in OPC cultured in the presence of D-Asp for 4 days (about 20%).

### D-Aspartate exposure stimulates axonal myelination and remyelination in cerebellar organotypic slice cultures

Next, to explore whether D-Asp may stimulate axonal myelination, organotypic cerebellar slices were incubated in the presence or in the absence of 100 μM D-Asp during 7 days *in vitro* (DIV; Fig 2A). Confocal immunofluorescence analysis for MBP and the axonal marker NF200 showed an increased axonal myelination in D-Asp-treated slices, as revealed by the significant upregulation of the myelination index compared to control slices (Fig 2B and C). In line, D-Asp treatment (100 μM) significantly upregulated MBP protein levels in exposed slices compared to control untreated slices (Fig 2D).

To investigate whether D-Asp treatment has functional significance for myelin repair, we evaluated its effects on remyelination after lysophosphatidylcholine (lysolecithin, LPC) exposure in cerebellar explants. To this aim, at 11 DIV, cerebellar organotypic slices were demyelinated with 0.5 mg/ml LPC (Sigma) for 15–17 h. Then, slices were treated at 2 days postlysolecithin (dpl) until 6 dpl or 10 dpl with 100 μM D-Asp or vehicle controls (Fig 2E). D-Asp treatment of demyelinated cerebellar organotypic slices significantly upregulated MBP protein levels at 6 dpl if compared to LPC-exposed slices (Fig 2G). D-Asp exposure significantly increased remyelination both at 6 dpl (Fig 2F–H) and at 10 dpl (Fig 2I and J) if compared to LPC-treated slices, as measured by remyelination index (colocalization of MBP and axonal neurofilament staining, normalized to area of neurofilament). These data demonstrate that D-Asp exposure accelerated remyelination *in vitro*.

### Blockade of NMDA receptors and NCX3 prevented D-Aspartate-induced myelin markers expression and intracellular $[Ca^{2+}]_i$ oscillations in oligodendrocyte precursors

The documented ability of D-Asp to activate NMDA receptors (Errico *et al*, 2008a,b; Krashia *et al*, 2016) and the significant role played by the activation of these ionotropic glutamate receptors and the $Na^+/Ca^{2+}$ exchanger NCX3 during oligodendrocyte maturation (Karadottir *et al*, 2005; Cavaliere *et al*, 2012; Boscia *et al*, 2013a; Friess *et al*, 2016) led us to investigate whether the boosting actions of D-Asp on OPC differentiation might involve NMDA receptors and NCX3 activation. To this aim, we first analyzed the effect of selective NMDA and NCX3 blockers on D-Asp-induced myelin markers increase, and then D-Asp effect on calcium signaling in OPC.

When MO3.13 progenitors were exposed for 3 days to 200 μM D-Asp in the presence of 10 μM (5S,10R)-(+)-5-methyl-10,11-dihydro-5H-dibenzo[a,d]cyclohepten-5,10-imine hydrogen maleate, (dizocilpine,

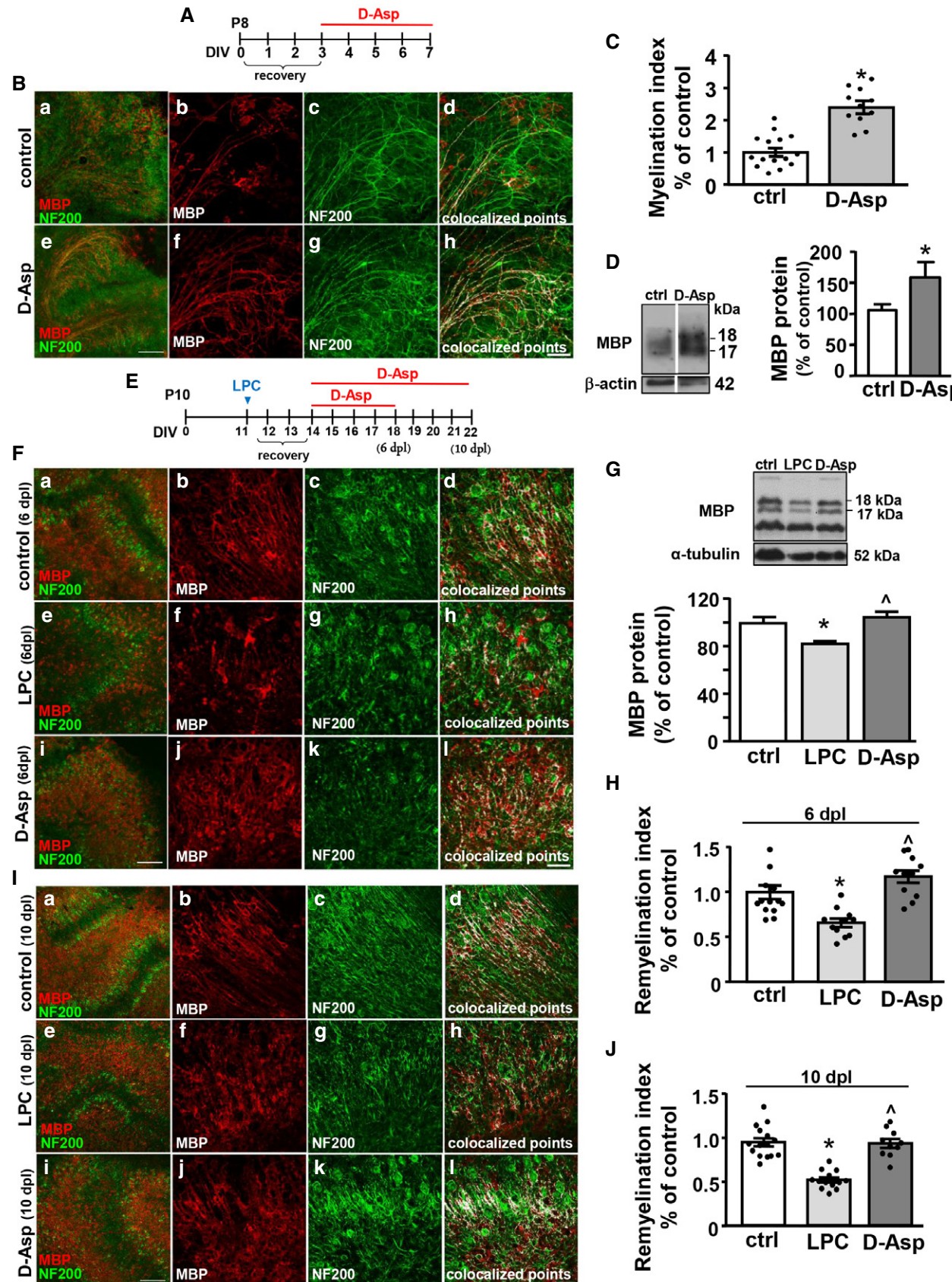

Figure 2.

◄

**Figure 2.  Effect of D-Asp treatment on myelination and remyelination in cerebellar organotypic slices.**

A  Schematic diagram showing D-Asp exposure protocol in organotypic slices.

B  Maximum intensity projection of *z*-stack confocal images displaying MBP (red) and NF200 (green) immunoreactivities in seven DIV cerebellar organotypic slices cultured in the absence (b–d) or in the presence of 100 μM D-Asp (f–h). Panels (d and h) display colocalized points (white). Panels (a and e) show representative low magnification images of seven DIV cerebellar slices cultured cultures in the absence (a) or in the presence of 100 μM D-Asp (e). Scale bars in (a and e): 200 μm; in (b–d) and (f–h): 50 μm.

C  Scatter plot histogram analysis of the myelination index in seven DIV cerebellar slices cultured in the absence or in the presence of 100 μM D-Asp.

D  Western blotting analysis of MBP protein levels from homogenates of organotypic cerebellar slices cultured in the absence or in the presence of 100 μM D-Asp. Data were normalized on the basis of β-actin and expressed as percentage of controls.

E  Schematic diagram showing D-Asp exposure protocol in organotypic slices after LPC exposure.

F  Maximum intensity projection of *z*-stack confocal images displaying MBP (red) and NF200 (green) immunoreactivities in control cerebellar organotypic slices (b–d), and in cerebellar slices at 6 days after LPC exposure (dpl) in the absence (f–h) or in the presence 100 μM D-Asp (j–l). Panels (d, h, and l) display colocalized points (white). Panels (a, e, and i) show representative low magnification images of seven DIV cerebellar slices cultured cultures in the absence (a) or in the presence of 100 μM D-Asp (e). Scale bars in (a, e, i): 200 μm; in (b–d, f–h, and j–l): 50 μm.

G  Western blotting and quantitative analysis of MBP protein levels from homogenates of organotypic cerebellar slices at 6 dpl cultured in the absence or in the presence of 100 μM D-Asp.

H  Scatter plot histogram analysis of the remyelination index in cerebellar explants at 6 dpl after LPC exposure cultured in the absence or in the presence of 100 μM D-Asp. Data were normalized to vehicle control.

I  Maximum intensity projection of *z*-stack confocal images displaying MBP (red) and NF200 (green) immunoreactivities in control cerebellar organotypic slices (b–d), and in cerebellar slices at 10 dpl in the absence (f–h) or in the presence of 100 μM D-Asp (j–l). Panels (d, h, and l) display colocalized points (white). Panels (a, e, and i) show representative low magnification images of seven DIV cerebellar slices cultured cultures in the absence (a) or in the presence of 100 μM D-Asp (e). Scale bars in (a, e, i): 200 μm; in (b–d, f–h, and j–l): 50 μm.

J  Scatter plot histogram analysis of the remyelination index in cerebellar explants at 10 dpl cultured in the absence or in the presence of 100 μM D-Asp. Data were normalized to vehicle control.

Data information: The values represent the means ± SEM. Level of significance was determined by using: in (C) two-tailed Student's *t*-test, \**P* < 0.05 versus control (*n* = 4 animals, 3–4 slices per group); (D) two-tailed Student's *t*-test, \**P* < 0.05 versus controls (*n* = 3); (G) one-way ANOVA *P* = 0.003 followed by Bonferroni *post hoc* test, \**P* < 0.05 versus control, ^*P* < 0.05 versus LPC (*n* = 3); (H) one-way ANOVA *P* < 0.001 followed by Bonferroni *post hoc* test, \**P* < 0.05 versus controls, ^*P* < 0.05 versus LPC (*n* = 4 animals, 3–4 slices per group); (J) one-way ANOVA *P* < 0.001 followed by Bonferroni *post hoc* test, \**P* < 0.05 versus controls, ^*P* < 0.05 versus LPC (*n* = 4 animals, 3–4 slices per group). See the exact *P*-values from comparisons tests in Appendix Table S2.

Source data are available online for this figure.

MK-801) or the NCX3 blockers, 30 nM N-(3-aminobenzyl)-6-{4-[(3-fluorobenzyl)oxy]phenoxy} nicotinamide (YM-244769) or 100 nM 5-amino-N-butyl-2-(4-ethoxyphenoxy)-benzamide hydrochloride (BED), the upregulation of both CNPase and MBP transcripts induced by D-Asp was significantly prevented (Fig 3A and B, respectively). In line, confocal studies indicated that D-Asp-exposed OPC, when cultured in the presence of MK-801 or NCX3 blockers for 3 days, displayed a significant higher number of immature neuron-glial antigen 2 NG2$^+$ cells, a lower number of mature MAG$^+$ oligodendrocytes compared to D-Asp-treated OPC (Fig 3C). By contrast, the number of double-labeled NG2$^+$-MAG$^+$ cells, presumably O4$^+$ cells, remained unchanged (data not shown). Interestingly, exposure of MO3.13 progenitors to 10–200 μM D-Asp significantly and dose-dependently upregulated NCX3 transcripts (Fig 3D), and this upregulation was significantly prevented by MK-801 (Fig 3E). By contrast, D-Asp incubation, either in the absence or in the presence of MK-801, did not influence NCX1 transcripts levels (Fig 3F). Exposure of oligodendrocyte progenitors with MK-801, YM-244769, or BED for 3 days did not affect neither NCX3 nor CNPase and MBP mRNA levels (Fig 3).

Then, we analyzed whether the activation of NMDA receptors and NCX3 exchangers might contribute to the changes in [Ca$^{2+}$]$_i$ following D-Asp treatment. Acute 100 μM D-Asp application induced an initial peak of [Ca$^{2+}$]$_i$ followed by an oscillatory Ca$^{2+}$ pattern both in human MO3.13 progenitors (~90%) and in the majority of rat primary OPC (~70%), as recorded by Fura-2 video imaging (Fig 4). Only a small percentage of OPC (~20%) underwent a single rise in [Ca$^{2+}$]$_i$ (data not shown).

Both the selective competitive and non-competitive NMDA receptor blockers, 2-amino-5-phosphonopentanoic acid (APV, 150 μM)

and MK-801 (10 μM), completely suppressed D-Asp-induced [Ca$^{2+}$]$_i$ oscillations both in MO3.13 and in rat primary OPC, but only partially affected the first [Ca$^{2+}$]$_i$ peak (Fig 4A and B). The [Ca$^{2+}$]$_i$ oscillation frequency in MO3.13 progenitors or primary OPC after D-Asp stimulation in the presence of MK-801 was 0.05 ± 0.007 Hz (MO3.13) and 0.1 ± 0.003 Hz (OPC), and in the presence of APV was 0.046 ± 0.007 (MO3.13) and 0.15 ± 0.002 (OPC), if compared to 0.2 ± 0.023 Hz (MO3.13) and 0.4 ± 0.005 Hz (OPC) calculated in the absence of MK-801.

Similarly, pharmacological blockade of NCX3 exchanger with either 30 nM YM-244769 or the most recently developed compound 100 nM BED (Secondo *et al*, 2015) completely suppressed D-Asp-induced [Ca$^{2+}$]$_i$ oscillations both in MO3.13 cells and in OPC (Fig 4C and D). The oscillation frequency in MO3.13 progenitors or OPC after D-Asp stimulation pretreated with BED was 0.013 ± 0.001 Hz and 0.2 ± 0.002 Hz, respectively, if compared to 0.2 ± 0.023 Hz (MO3.13) and 0.4 ± 0.005 Hz (OPC) calculated in the absence of BED. The oscillation frequency in MO3.13 progenitors or primary OPC pretreated with YM-244769 was 0.004 ± 0.001 Hz and 0.1 ± 0.002 Hz, respectively, if compared to 0.2 ± 0.023 Hz (MO3.13) or 0.4 ± 0.005 Hz (OPC) calculated in the absence of YM-244769. Nevertheless, both YM-244769 and BED only partially prevented the initial rise of [Ca$^{2+}$]$_i$ observed following D-Asp application (Fig 4C, D and c, d).

To further investigate the contribution of NCX3 exchanger to D-Asp-induced [Ca$^{2+}$]$_i$ oscillations, we recorded calcium response in MO3.13 progenitors previously silenced for *ncx3* gene or in mouse OPC obtained from *ncx3*$^{+/+}$ and *ncx3*$^{-/-}$ mice. As shown in Fig 4E and F, quantitative analysis of the oscillatory

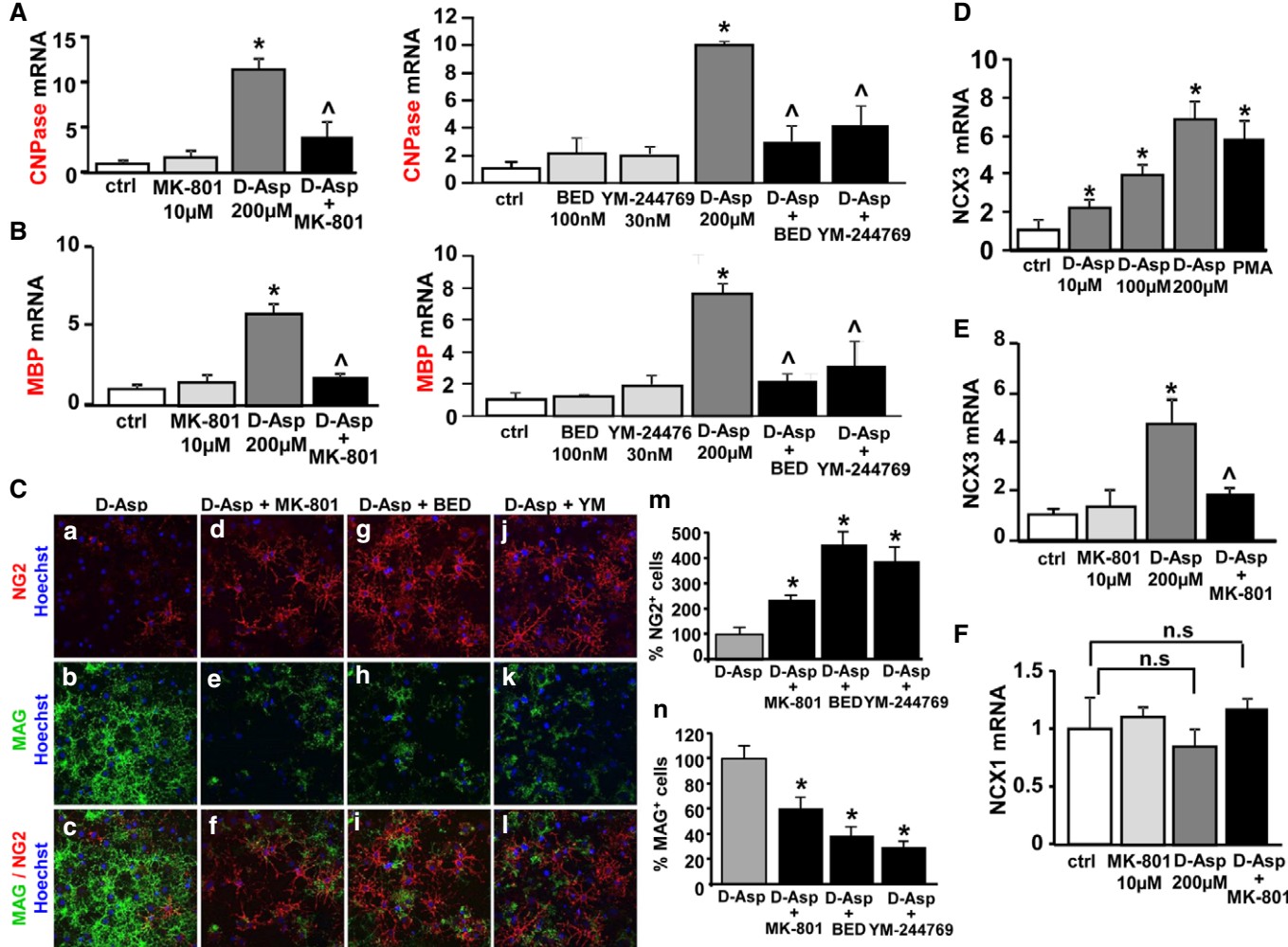

**Figure 3.  Effect of the NMDA receptors antagonist and NCX3 blockers on D-Asp-induced myelin marker expression.**

A    RT–PCR of CNPase mRNA expression in oligodendrocyte MO3.13 progenitors under control conditions and following 200 μM D-Asp exposure for 3 days, in the absence or in the presence of 10 μM MK-801 (left panel), or 30 nM YM-244769 or 100 nM BED (right panel). Graphs show quantification of ratio of CNPase to L19.

B    RT–PCR of MBP mRNA expression in MO3.13 cells under control conditions and following D-Asp exposure for 3 days, in the absence or in the presence of 10 μM MK-801 (left panel), or 30 nM YM-244769 or 100 nM BED (right panel) Graphs show quantification of ratio of MBP to L19.

C    (a–l) Confocal images displaying the expression of NG2 and MAG proteins in rat primary OPC cultured in the presence of D-Asp for 3 days, in the absence (a–c) or in the presence of 10 μM MK-801 (d–f), or 30 nM YM-244769 (g–i) or 100 nM BED (j–l). (m, n) Quantitative analysis of NG2+ and MAG+ cells in rat primary OPC cultured in the presence of D-Asp for 3 days, in the absence or in the presence of 10 μM MK-801, 30 nM YM-244769 or 100 nM BED.

D    RT–PCR of NCX3 mRNA expression under control conditions and following 10–200 μM D-Asp exposure or 100 nM PMA for 3 days.

E    RT–PCR of NCX3 mRNA expression following 200 μM D-Asp exposure, in the absence or in the presence of 10 μM MK-801.

F    RT–PCR of NCX1 mRNA expression following 200 μM D-Asp exposure, in the absence or in the presence of 10 μM MK-801. Graphs show quantification of ratio of NCX1 and NCX3 to L19.

Data information: The values represent the means ± SEM from three independent experimental sessions. Level of significance was determined by using: in (A) left, one-way ANOVA $P = 0.009$ followed by Tukey's *post hoc* test, *$P < 0.05$ versus control, ^$P < 0.05$ versus D-Asp ($n = 3$); (A) right, one-way ANOVA $P = 0.0001$ followed by Tukey's *post hoc* test, *$P < 0.05$ versus control, ^$P < 0.05$ versus D-Asp ($n = 3$); (B) left, one-way ANOVA $P = 0.0001$ followed by Tukey's *post hoc* test, *$P < 0.05$ versus control, ^$P < 0.05$ versus D-Asp ($n = 3$); (B) right, one-way ANOVA $P = 0.0004$ followed by Tukey's *post hoc* test, *$P < 0.05$ versus control, ^$P < 0.05$ versus D-Asp ($n = 3$); (C, m and C, n) one-way ANOVA $P < 0.0001$ followed by Bonferroni *post hoc* test, *$P < 0.05$ versus control, ^$P < 0.05$ versus D-Asp ($n = 3$); (D) one-way ANOVA $P = 0.0001$ followed by Tukey's *post hoc* test, *$P < 0.05$ versus control, ^$P < 0.05$ versus D-Asp ($n = 3$); (E) one-way ANOVA $P = 0.0083$ followed by Tukey's *post hoc* test, *$P < 0.05$ versus control, ^$P < 0.05$ versus D-Asp ($n = 3$), n.s, not significant. See the exact $P$-values from comparisons tests in Appendix Table S3.

index revealed that silencing or knocking out *ncx3* gene significantly suppressed the $[Ca^{2+}]_i$ oscillating pattern following D-Asp exposure. Similarly to what we observed by using pharmacological approach, blocking NCX3 activity by using silencing or transgenic approaches only partially affected the initial $[Ca^{2+}]_i$ peak following D-Asp application (Fig 4E and F).

**Blockade of glutamate transporters and AMPA receptors prevented D-Asp-elicited inward currents and the initial $[Ca^{2+}]_i$ rise after D-Asp stimulation in oligodendrocyte precursors**

Based on the emerging role of AMPA receptors during OPC development (Fannon *et al*, 2015; Gautier *et al*, 2015;

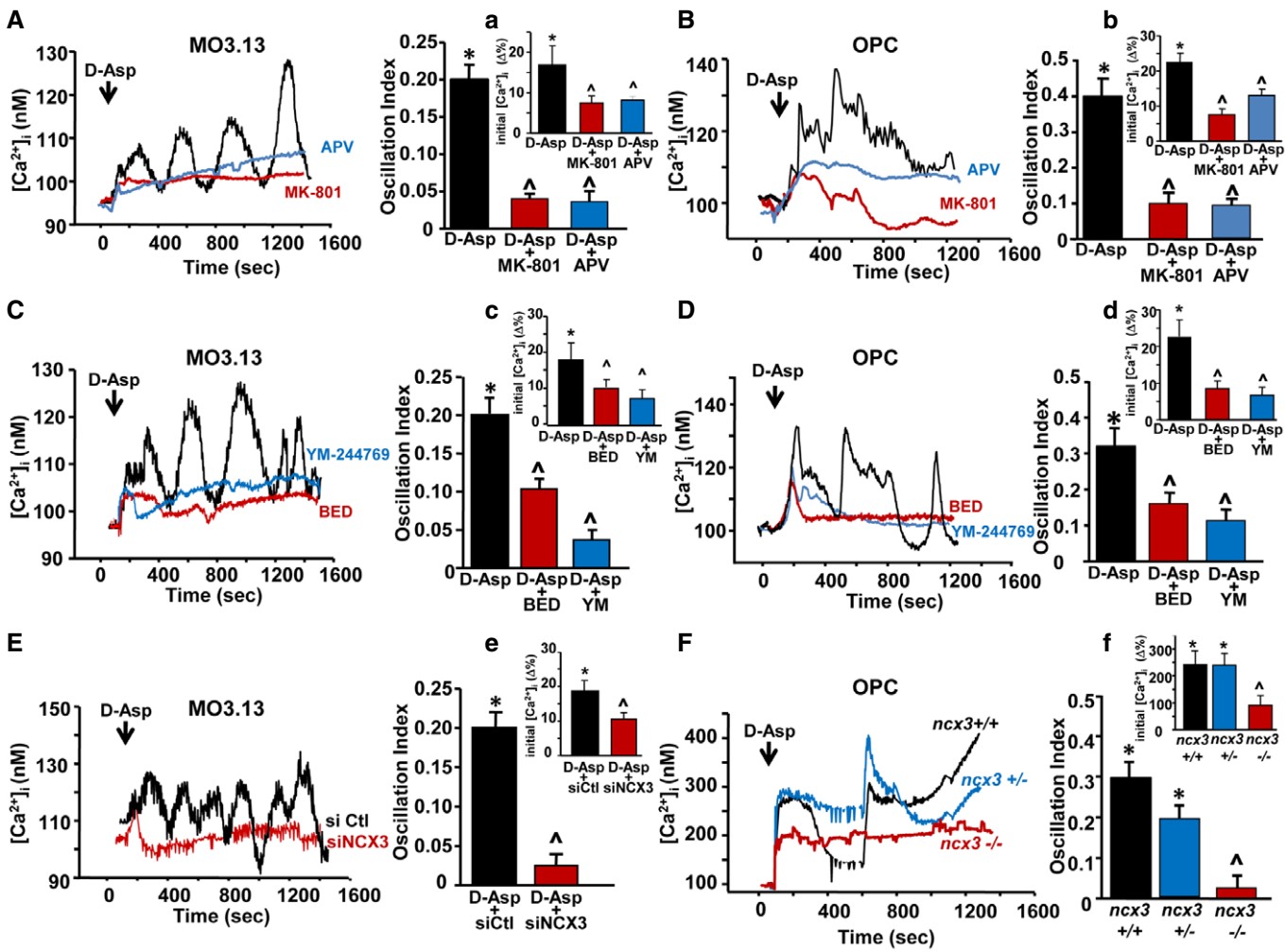

**Figure 4. Contribution of NMDA receptor and NCX3 exchanger activities on D-Asp-induced [Ca$^{2+}$]$_i$ levels in oligodendrocyte precursors.**

A, B    Left panels: Superimposed single-cell traces representative of the effect of 100 μM D-Asp on [Ca$^{2+}$]$_i$ detected in MO3.13 cells (A) and primary OPC (B) in the absence or in the presence of 10 μM MK-801 or 150 μM APV. Right panels: Quantification of the oscillation index in MO3.13 cells (A) and primary OPC (B) in the absence or in the presence of 10 μM MK-801 or 150 μM APV. (a, b) Quantification of the initial [Ca$^{2+}$]$_i$ increase elicited by D-Asp measured as Δ% of peak versus basal values in the absence or in the presence of 10 μM MK-801 or 150 μM APV, both in MO3.13 cells (a) and in primary OPC (b). MK-801 and APV were preincubated 10 min before registration.

C, D    Left panels: superimposed single-cell traces representative of the effect of 100 μM D-Asp on [Ca$^{2+}$]$_i$ detected in MO3.13 cells (C) and primary OPC (D) in the absence or in the presence of 30 nM YM-244769 or 100 nM BED. Right panels: quantification of the oscillation index in MO3.13 cells (C) and primary OPC (D) in the absence or in the presence of 30 nM YM-244769 or 100 nM BED. (c, d) Quantification of the initial [Ca$^{2+}$]$_i$ increase elicited by D-Asp measured as Δ% of peak versus basal values in the absence or in the presence of 30 nM YM-244769 or 100 nM BED, both in MO3.13 cells (c) and in primary OPC (d). YM-244769 or BED was preincubated 10 min before registration.

E    Left: Superimposed single-cell traces representative of the effect of 100 μM D-Asp on [Ca$^{2+}$]$_i$ detected in MO3.13 cells in the presence of siCtl or sincx3 silencing. Right: Quantification of the oscillation index in MO3.13 cells in the absence or in the presence of sincx3. (e) Quantification of the initial [Ca$^{2+}$]$_i$ increase elicited by D-Asp and measured as Δ% of peak versus basal values in the absence or in the presence of sincx3.

F    Left: Superimposed single-cell traces representative of the effect of 100 μM D-Asp on [Ca$^{2+}$]$_i$ detected in primary OPC obtained from wild-type ncx3$^{+/+}$, heterozygous ncx3$^{+/-}$, and knockout ncx3$^{-/-}$ mice. Right: Quantification of the oscillation index elicited by D-Asp in primary mouse OPC obtained from ncx3$^{+/+}$, ncx3$^{+/-}$, and ncx3$^{-/-}$ mice. (f) Quantification of the initial [Ca$^{2+}$]$_i$ increase measured as Δ% of peak versus basal values.

Data information: The values represent the mean ± SEM from three independent experimental sessions. Level of significance was determined by using: in (A and a) one-way ANOVA $P < 0.001$ followed by Bonferroni *post hoc* test, *$P < 0.05$ versus control (basal value), ^$P < 0.05$ versus D-Asp. Data are reported as mean of 23–30 cells in each group, $n = 3$ biological replicates; (B and b) one-way ANOVA, $P < 0.001$ followed by Bonferroni *post hoc* test, *$P < 0.05$ versus control (basal value), ^$P < 0.05$ versus D-Asp. Data are reported as mean of 20–23 cells in each group, $n = 3$ biological replicates; (C and c) one-way ANOVA $P < 0.001$ followed by Bonferroni *post hoc* test, *$P < 0.05$ versus control (basal value), ^$P < 0.05$ versus D-Asp. Data are reported as mean of 19–30 cells in each group, $n = 3$ biological replicates; (D and d) one-way ANOVA $P < 0.001$ followed by Bonferroni *post hoc* test, *$P < 0.05$ versus control (basal value), ^$P < 0.05$ versus D-Asp. Data are reported as mean of 18–20 cells in each group, $n = 3$ biological replicates; (E and e) one-way ANOVA $P < 0.001$ followed by Bonferroni *post hoc* test, *$P < 0.05$ versus sictl, ^$P < 0.05$ versus D-Asp + sictl. Data are reported as mean of 25–30 cells in each group, $n = 3$ biological replicates; (F and f) one-way ANOVA $P < 0.001$ and $P = 0.003$, respectively, followed by Bonferroni *post hoc* test, *$P < 0.05$ versus basal value, ^$P < 0.05$ versus ncx3$^{+/+}$ and ncx3$^{+/-}$. Data are reported as mean of 10–19 cells in each group, $n = 3$ biological replicates. See the exact $P$-values from comparisons tests in Appendix Table S4.

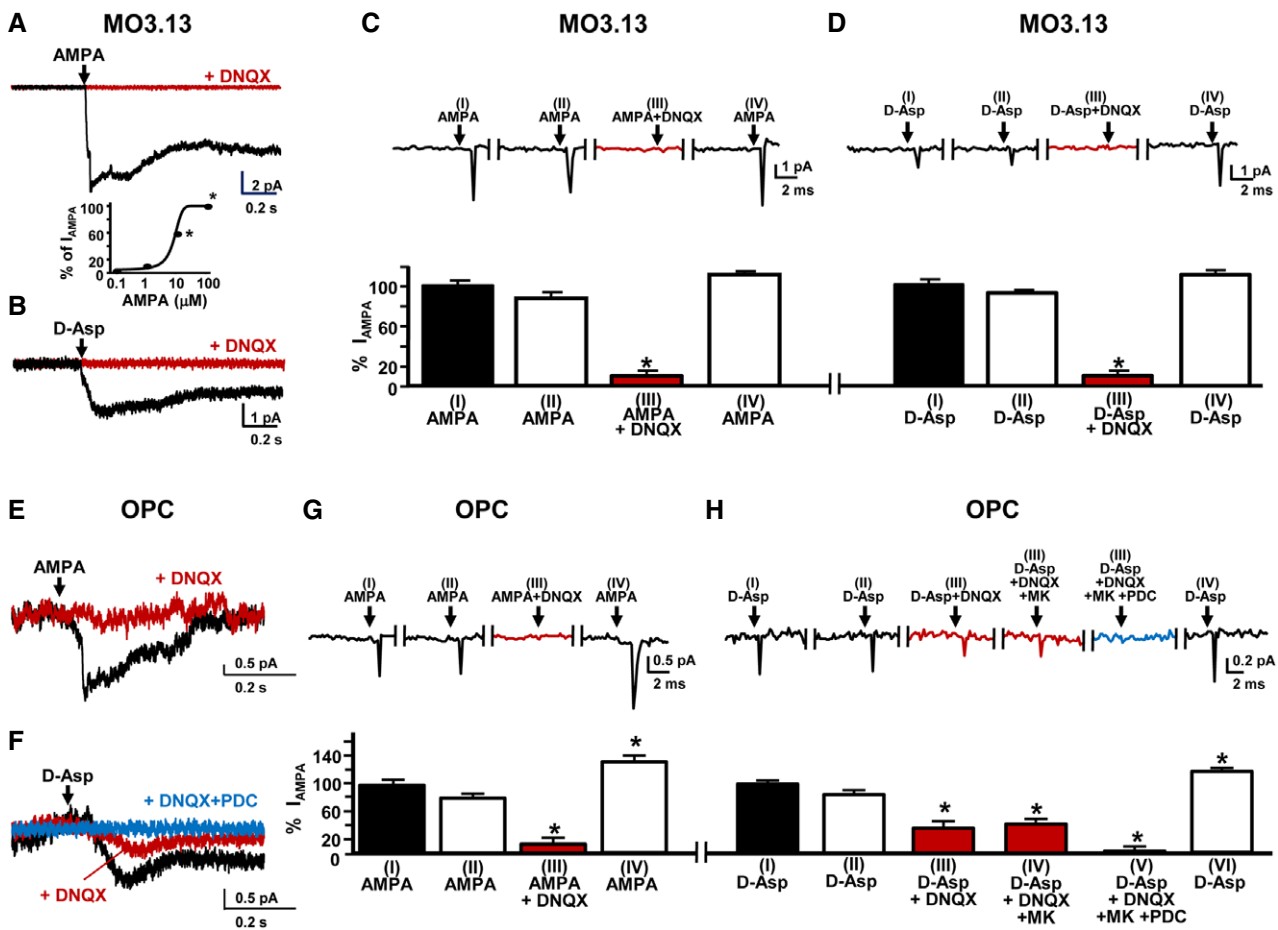

**Figure 5. Effect of AMPA receptors and glutamate transporters blockade on D-Asp elicited inward currents in oligodendrocyte precursors.**

A   Representative inward current traces elicited by 10 μM AMPA in human oligodendrocyte MO3.13 progenitor cells in the absence or in the presence of 10 μM DNQX. The concentration-dependent curve of AMPA (1–100 μM) on inward currents is showed at the bottom.

B   Representative inward current traces elicited by 100 μM D-Asp application in MO3.13 cells in the absence or in the presence of 10 μM DNQX.

C   Representative traces of the inward current (upper panel) and the corresponding quantification (bottom panel) observed in human oligodendrocyte MO3.13 progenitors after: (I) the first application of 10 μM AMPA alone; (II) the second application of 10 μM AMPA alone; (III) the third application of 10 μM AMPA in the presence of 10 μM DNQX; and (IV) the fourth application of 10 μM AMPA alone. Data are expressed as percentage of control.

D   Representative traces of the inward current (upper panel) and the corresponding quantification (bottom panel) observed in human oligodendrocyte MO3.13 progenitors after: (I) the first application of 100 μM D-Asp alone; (II) the second application of 100 μM D-Asp alone; (III) the third application of 100 μM D-Asp in the presence of 10 μM DNQX; and (IV) the fourth application of 100 μM D-Asp alone. Data are expressed as percentage of control.

E   Representative inward current traces elicited by 10 μM AMPA in rat primary OPC in the absence or in the presence of 10 μM DNQX.

F   Representative inward current traces elicited by 100 μM D-Asp in rat primary OPC in the absence or in the presence of 10 μM DNQX or 10 μM DNQX + 20 μM PDC.

G   Representative traces of the inward current (upper panel) and the corresponding quantification (lower panel) observed in rat primary OPC after: (I) the first application of 10 μM AMPA alone; (II) the second application of 10 μM AMPA alone; (III) the third application of 10 μM AMPA in the presence of 10 μM DNQX; and (IV) the fourth application of 10 μM AMPA alone. Data are expressed as percentage of control.

H   Representative traces of the inward current (upper panel) and the corresponding quantification (lower panel) observed in rat primary OPC after: (I) the first application of 100 μM D-Asp alone; (II) the second application of 100 μM D-Asp alone; (III) the third application of 100 μM D-Asp in the presence of 10 μM DNQX; (IV) the fourth application of 100 μM D-Asp in the presence of 10 μM DNQX + 10 μM MK801; (V) the fifth application of 100 μM D-Asp in the presence of 10 μM DNQX + 10 μM MK-801 + 20 μM PDC; and (VI) the sixth application of 100 μM D-Asp alone. Data are expressed as percentage of control.

Data information: The values represent the mean ± SEM from three independent experimental sessions. Level of significance was determined by using: in (A) one-way ANOVA $P < 0.001$ followed by Bonferroni *post hoc* test, *$P < 0.05$ versus 1 μM AMPA, $n = 3$ biological replicates; (C and G) one-way ANOVA $P < 0.001$ followed by Bonferroni *post hoc* test, *$P < 0.05$ versus AMPA(I). Data are reported as mean of at least 25 cells in each group, $n = 3$ biological replicates; (D and H) one-way ANOVA $P < 0.001$ followed by Bonferroni *post hoc* test, *$P < 0.05$ versus D-Asp (I). Data are reported as mean of at least 25 cells for each group, $n = 3$ biological replicates. See the exact $P$-values from comparisons tests in Appendix Table S5.

Kougioumtzidou *et al*, 2017), we investigated whether D-Asp effects on oligodendrocytes may also involve AMPA receptor activation.

Electrophysiological experiments performed by patch-clamp in whole-cell configuration showed that when MO3.13 cells were clamped at −70 mV, 1–100 μM AMPA elicited a concentration-dependent

inward current which was prevented by the AMPA receptor antagonist 10 μM 6,7 dinitroquinoxaline-2,3-dione (DNQX; Fig 5A and C). Similarly, 100 μM AMPA elicited an inward current that was completely blocked by 10 μM DNQX in rat primary OPC (Fig 5E and G). In the same experimental conditions, D-Asp elicited an inward current that was completely blocked by 10 μM DNQX in MO3.13 progenitors (Fig 5B and D), but not in rat primary OPC (Fig 5F and H).

Both AMPA and D-Asp were able to elicit comparable inward currents after two consecutive applications both in MO3.13 cells (Fig 5C and D) and in rat primary OPC (Fig 5G and H), thus excluding the possibility that the desensitization of AMPA receptors might contribute to the inhibitory effects observed after DNQX incubation. Moreover, the inhibition exerted by DNQX was reversible, since both AMPA and D-Asp were able to elicit again the inward current after the washout (Fig 5C, D, G, and H). Next, we investigated whether NMDA receptor or glutamate transporter activation contributed to D-Asp-induced inward currents in OPC. Interestingly, D-Asp residual inward currents in OPC were unaltered by the application of the NMDA receptor antagonist MK-801 (10 μM), but completely inhibited by the glutamate transporter blocker L-*trans*-pyrrolidine-2,4-dicarboxylic acid (PDC; 20 μM; Fig 5F and H). These results suggest that the activation of both AMPA receptors and the glutamate transporters contributes to D-Asp-induced inward currents in OPC.

Based on these results, we investigated whether the activation of AMPA receptors might be involved in the initial $[Ca^{2+}]_i$ peak elicited by 100 μM D-Asp stimulation in oligodendrocyte precursors. As shown in Fig EV1, 10 μM AMPA perfusion triggered $[Ca^{2+}]_i$ oscillations in MO3.13 oligodendrocyte progenitors. The selective and competitive AMPA receptor antagonist DNQX was preincubated with AMPA for 10 min to block the channel in a use-dependent way before AMPA stimulation. Under these experimental conditions, no significant changes in $[Ca^{2+}]_i$ were recorded after 10 μM AMPA addition (Fig EV1A). Interestingly, in oligodendrocyte MO3.13 progenitors pretreated with DNQX + AMPA, D-Asp failed to induce

the first $[Ca^{2+}]_i$ peak (Fig EV1B and b) as well as $[Ca^{2+}]_i$ oscillation pattern (Fig EV1B). By contrast, in rat primary OPC, the pharmacological blocking of AMPA receptors with 1.5 μM cyanquixaline (6-cyano-7-nitroquinixaline-2,3-dione, CNQX; data not shown) or of both AMPA and NMDA receptors with 25 μM CNQX completely abolished the $[Ca^{2+}]_i$ oscillation pattern induced by D-Asp (Fig EV1C), but did not fully prevent the initial $[Ca^{2+}]_i$ rise (Fig EV1C and c). In line with electrophysiological findings performed in primary OPC, the glutamate transporter blocker PDC (20 μM) fully prevented the initial $[Ca^{2+}]_i$ peak (Figs 6, and EV1D and d) as well as $[Ca^{2+}]_i$ oscillation pattern after D-Asp exposure (Fig EV1D).

## D-Aspartate treatment improves motor performance, attenuates demyelination, and accelerates remyelination in the cuprizone mouse model

To explore the action of D-Asp treatment during acute demyelination and remyelination, we analyzed the effect of D-Asp treatment *in vivo,* in mice fed with the copper chelator cuprizone. To analyze the effects on acute demyelination, 20 mM D-Asp was given to the mice in drinking solution for 5 weeks during all the cuprizone treatment (demyelination, DEM). To explore the effects on remyelination (remyelination, REM), D-Asp treatment was initiated 1 week before cuprizone withdrawal (D-Asp II) or after cuprizone withdrawal (D-Asp I) and maintained for additional 2.5 weeks (Fig 6A). The effects of D-Asp treatment during demyelination and remyelination were assessed on motor coordination performance in beam balance and both fixed-speed and accelerating rotarod test (Figs 6 and EV2). Mice completed three trials for each task during 1, 3 and 5 weeks of demyelination and after 2.5 weeks of remyelination. The number of falls and the latencies across daily trials and their average are calculated and shown in Figs 6 and EV2. One week after cuprizone feeding, no significant changes in motor performance among animal groups were assessed with both beam balance and rotarod tests

---

**Figure 6. Effects of D-Asp treatment on motor coordination performance in beam balance and rotarod test.**

A   Schematic representation of the administration protocols for D-Asp treatment in mice. Cuprizone was administered for weeks 1–5 (blue) to induce demyelination (DEM) in the *corpus callosum*. Then, mice returned to normal diet for weeks 6–7.5 (red), allowing recovery and remyelination (REM). D-Asp (20 mM in drinking solution) was given to the mice for 5 weeks concomitantly the cuprizone treatment (DEM); D-Asp treatment was initiated after cuprizone withdrawal and maintained for additional 2.5 weeks (D-Asp I); D-Asp treatment was initiated 1 week before cuprizone withdrawal and maintained for additional 2.5 weeks (D-Asp II).

B, C   Beam crossing latency during daily training (left panel, averaged across three trials per day) and average latency to cross the beam over 3 consecutive days (right panel), recorded in control (open circles), cuprizone- (filled black circles), and cuprizone + D-Asp-treated mice (filled red triangles) during 3 weeks (B) and 5 weeks (C) of cuprizone feeding.

D   Beam crossing latency during daily training (left panel, averaged across three trials per day) and average latency to cross the beam over 3 consecutive days (middle and right panels) recorded after 2.5 weeks of remyelination in control (open circles), vehicle- (filled black circles), and D-Asp-treated mice. D-Asp (I) (filled gray triangles) refers to the group of mice which received D-Asp only for 2.5 weeks after cuprizone withdrawal; D-Asp (II) (filled red triangles) refers to the group of mice which received D-Asp during the last week of cuprizone feeding and for 2.5 additional weeks after cuprizone withdrawal.

E, F   Latency to fall during daily training in accelerating rotarod test (left panel, averaged across three trials per day) and average over 3 consecutive days (right panel) recorded in control (open circles), cuprizone- (filled black circles), and D-Asp-treated (filled red triangles) mice at 3 weeks (E) and 5 weeks (F) of cuprizone feeding.

G   Latency to fall during daily training in accelerating rotarod test (left panel, averaged across three trials per day) and average over 3 consecutive days (right panel) recorded after 2.5 weeks of remyelination (REM) in control, vehicle-, and D-Asp-treated mice. D-Asp (II) refers to the group of mice which received D-Asp during the last week of cuprizone feeding and 2.5 additional weeks after cuprizone withdrawal.

Data information: The values represent the means ± SEM ($n$ = 8 mice for each group). Level of significance was determined by using: in (B–G) left panels (daily training), for each day (d) one-way ANOVA $P = 0.0013$ (B, 1 day), $P = 0.001$ (C, 1 day), $P = 0.0027$ (D, 3 day), $P = 0.0025$ (D, 2 day), $P < 0.0001$ (E, 3 day), $P = 0.0082$ (F, 3 day), $P < 0.0001$ (G, 1 day), $P = 0.0044$ (G, 2 day), $P = 0.0038$ (G, 3 day), followed by Bonferroni *post hoc* test, [#]$P < 0.05$ versus cpz at day 1, [§]$P < 0.05$ versus cpz at day 2, *$P < 0.05$ versus cpz at day 3. In B–G, right panels (average): one-way ANOVA $P = 0.0001$ (B), $P < 0.0001$ (C), $P < 0.0001$ (D, for both middle and right panels), $P = 0.0002$ (E), $P < 0.0001$ (F), $P < 0.0001$ (G), followed by Bonferroni *post hoc* test *$P < 0.05$ versus control; [^]$P < 0.05$ versus cpz or vehicle (veh). See the exact $P$-values from comparisons tests in Appendix Table S6.

(data not shown). By contrast, after 3 and more so after 5 weeks, cuprizone mice displayed a significant increase in latency to transverse the beam and number of falls from the rotarod, and a shorter latency to fall off the accelerating rotarod if compared to control mice (Figs 6 and EV2). Interestingly, cuprizone mice treated with D-Asp performed significantly better and the performance often improved over trials when compared to cuprizone group during 3 and 5 weeks of demyelination. In particular, D-Asp mice showed significantly shorter latency to walk the beam, reduced number of falls from the rotarod, and increased latencies to fall off the accelerating rotarod, when compared to cuprizone group (Figs 6 and EV2).

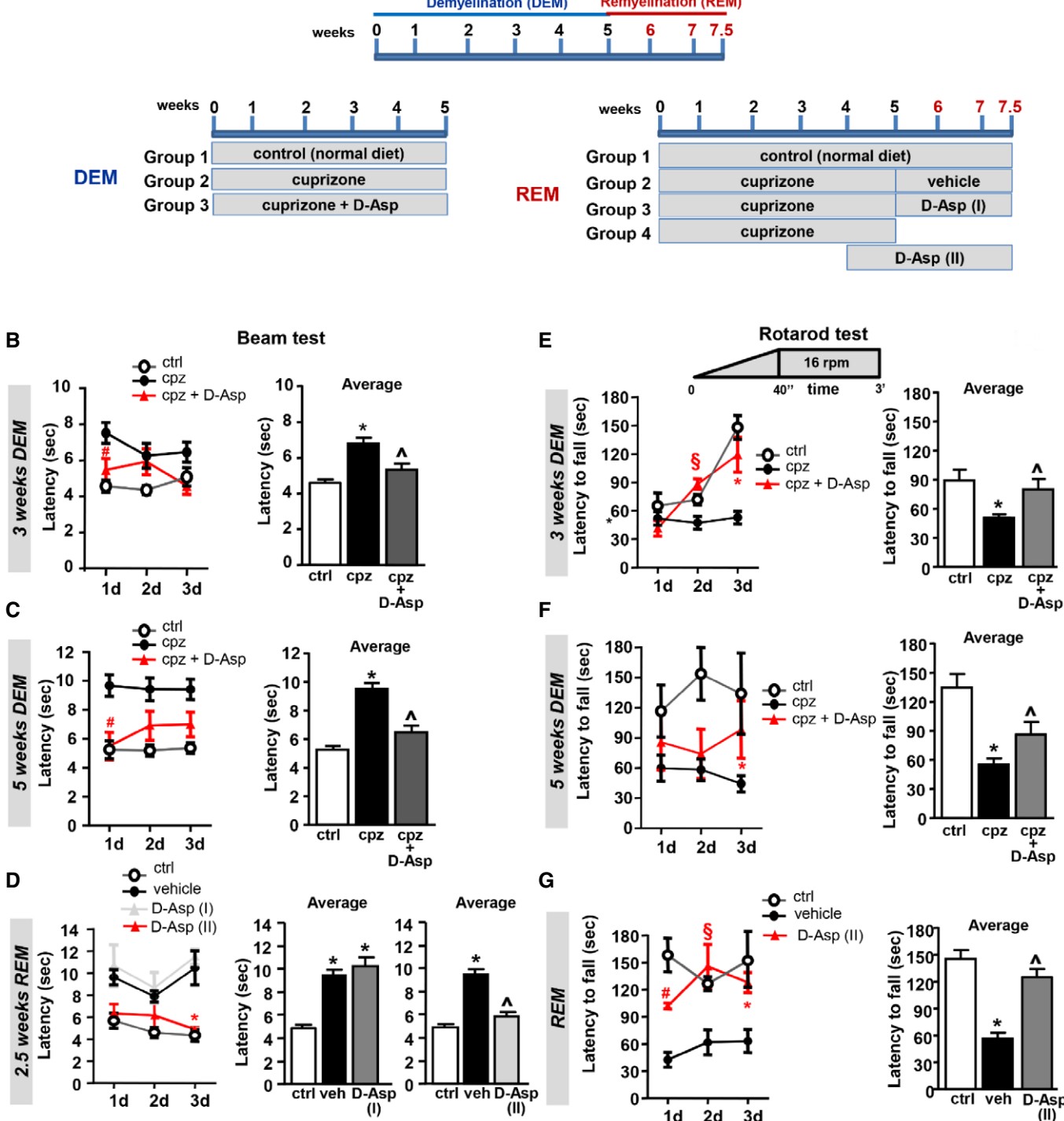

Figure 6.

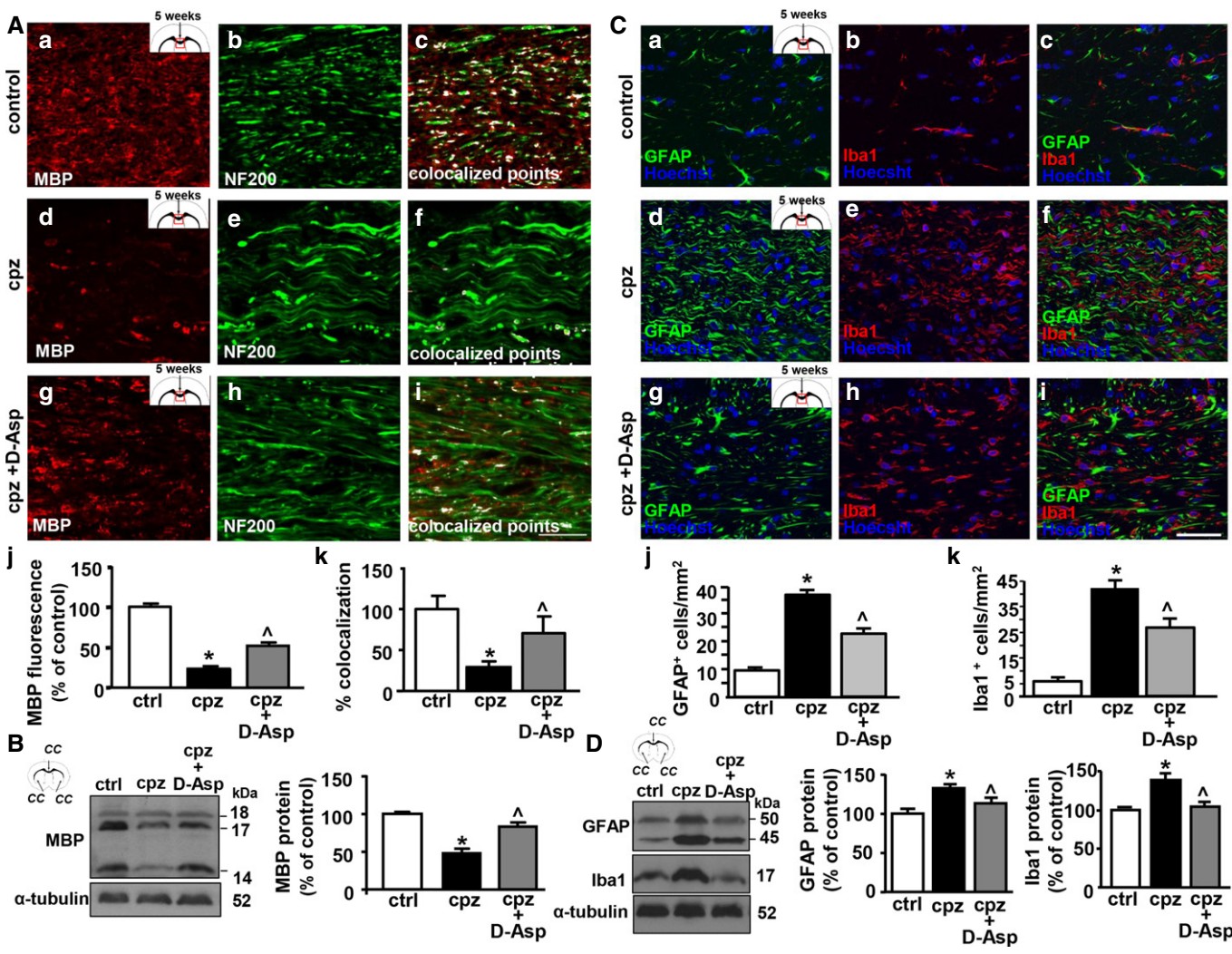

**Figure 7. Effects of D-Aspartate treatment on *corpus callosum* demyelination.**

A   Representative confocal double immunofluorescence images displaying MBP (red) and NF200 (green) distribution and their coexpression (white) in the middle *corpus callosum* of septostriatal sections of control mice (a–c), and of mice fed with cuprizone for 5 weeks in the absence (d–f) or in the presence of D-Asp (g–i). Scale bars in (a–i): 20 μm. (j), Densitometric analysis of MBP immunofluorescence signal and (k) quantification of MBP-NF200 colocalized points in the middle *corpus callosum* of septostriatal sections from control mice, and from mice fed with cuprizone for 5 weeks in the absence or in the presence of D-Asp.

B   Western blot and densitometric analysis of MBP levels in *corpus callosum* lysates from control and cuprizone-treated mice for 5 weeks in the absence or in the presence of D-Asp, respectively. Data were normalized on the basis of α-tubulin and expressed as percentage of controls.

C   Representative confocal double immunofluorescence images displaying GFAP (green) and Iba1 (red) distribution in the middle *corpus callosum* of septostriatal sections of control mice (a–c) and of mice fed with cuprizone for 5 weeks in the absence (d–f) or in the presence of D-Asp (g–i). Scale bars in (a–i): 50 μm. (j, k) Quantification of GFAP[+] and Iba1[+] cells in the middle *corpus callosum* of septostriatal sections of control mice and of mice fed with cuprizone for 5 weeks in the absence or in the presence of D-Asp. Data were normalized to the total cell number (Hoechst signal) and expressed as percentage of controls.

D   Western blot (left panel) and densitometric analyses (middle and right panels) of GFAP and Iba1 protein levels in *corpus callosum* lysates obtained from control or cuprizone-treated mice for 5 weeks in the absence or in the presence of D-Asp.

Data information: The values represent the means ± SEM. Level of significance was determined by using in: (A, j) one-way ANOVA *P* < 0.0001 followed by Bonferroni *post hoc* test, \**P* < 0.05 versus control, ^*P* < 0.05 versus cpz (*n* = 4 mice for each group); (A, k) one-way ANOVA *P* = 0.0162 followed by Newman–Keuls *post hoc* test, \**P* < 0.05 versus control, ^*P* < 0.05 versus cpz (*n* = 4 mice for each group); (B) one-way ANOVA *P* < 0.0001 followed by Bonferroni *post hoc* test, \**P* < 0.05 versus control, ^*P* < 0.05 versus cpz (*n* = 3 mice for each group); (C, j, k) one-way ANOVA *P* < 0.0001 followed by Bonferroni *post hoc* test, \**P* < 0.05 versus control, ^*P* < 0.05 versus cpz (*n* = 4 mice for each group); (D) middle panel, one-way ANOVA *P* = 0.0146 followed by Bonferroni *post hoc* test, \**P* < 0.05 versus control, ^*P* < 0.05 versus cpz (*n* = 3 mice for each group); (D) right panel, one-way ANOVA *P* = 0.0026 followed by Bonferroni *post hoc* test, \**P* < 0.05 versus control, ^*P* < 0.05 versus cpz (*n* = 3 mice for each group). See the exact *P*-values from comparisons tests in Appendix Table S7.
Source data are available online for this figure.

Assessment of motor performance during remyelination revealed that mice treated with D-Asp 1 week before cuprizone withdrawal and for 2.5 weeks after cuprizone withdrawal (D-Asp-II) presented improved motor performance when compared to cuprizone mice. By contrast, no significant improvement in motor skills was observed when D-Asp was given immediately after cuprizone

withdrawal (D-Asp I). No difference in overall performance across days was observed between control groups on beam balance and rotarod tests (Figs 6 and EV2).

Next, we investigated the effects of D-Asp treatment on myelin loss and inflammation 5 weeks after cuprizone-induced demyelination. Quantitative confocal immunofluorescence analysis performed in the middle *corpus callosum* showed a significant reduction in MBP fluorescence intensity in cuprizone mice when compared to control or D-Asp-treated mice (Fig 7A). Furthermore, coexpression analysis performed with anti-MBP and anti-NF200 antibodies in the middle *corpus callosum* revealed that the percentage of colocalization between MBP and NF200 was reduced after cuprizone treatment, but it was significantly preserved by D-Asp treatment (Fig 7A). In accordance, immunoblot analysis of *corpus callosum* lysates revealed a significant reduction in MBP protein levels after cuprizone treatment for 5 weeks, if compared to untreated controls. This reduction was significantly prevented by D-Asp treatment, as revealed by the quantitative analysis of MBP immunoreactive bands (Fig 7B). Furthermore, cell quantification in the middle *corpus callosum* showed a significant reduced number of glial fibrillary acidic protein GFAP$^+$ astrocytes and ionized calcium-binding adapter molecule 1 Iba1$^+$ microglia in D-Asp-treated group (Fig 7C), thus suggesting that D-Asp treatment also partially prevented demyelination-associated inflammation following cuprizone treatment. In line, Western blot analysis revealed that the protein levels of both GFAP and Iba1 were more intensely detected in *corpus callosum* homogenates of cuprizone-treated mice if compared to D-Asp-treated mice (Fig 7D).

Next, to investigate the effects of D-Asp on oligodendrocyte survival upon cuprizone treatment and on oligodendrocyte maturation during *corpus callosum* remyelination we analyzed the number of cells immunostained for Olig2, a transcription factor expressed in all cell types of the oligodendrocyte lineage, and the number of Olig2$^+$ cells coexpressing the adenomatous polyposis coli CC1, a marker of mature oligodendrocytes. Quantitative colocalization experiments performed in the *corpus callosum* of cuprizone-treated mice, in the absence or in the presence of D-Asp for 5 weeks, showed that the number of both Olig2$^+$ cells and those coexpressing CC1 (Olig2$^+$/CC1$^+$ cells) was significantly higher in D-Asp-treated mice compared to cuprizone-fed animals (Fig 8A). This suggests that D-Asp treatment during demyelination has a protective role on oligodendrocytes, thus explaining its attenuating effects on myelin damage. In accordance with confocal studies, quantitative ultrastructural electron microscopy (EM) analysis on the middle *corpus callosum* revealed that the percentage of myelinated axons at 5 weeks of demyelination was significantly higher in D-Asp-treated group (Fig 8B and C).

Confocal quantitative double immunofluorescence experiments performed in the middle *corpus callosum* 2.5 weeks after cuprizone withdrawal showed that the number of both Olig2$^+$ cells and Olig2$^+$-CC1$^+$ cells was significantly higher in mice treated with D-Asp compared to cuprizone-fed animals (vehicle, Fig 8A). Moreover, quantitative ultrastructural analysis performed 2.5 weeks after cuprizone withdrawal showed that D-Asp(II) treatment accelerates remyelination in the *corpus callosum*, as revealed by the significant lower G-ratio values of myelinated axons in D-Asp-treated mice compared to vehicle mice (Fig 8B, D, and E). Interestingly, D-Asp treatment during remyelination significantly increased the

percentage of small-diameter myelinated axons (0.2–0.4 μm) compared to vehicle mice, whereas axons with large diameter (≥ 0.5 μm) were unaffected (Fig 8F).

## Discussion

The present study shows that D-Aspartate treatment stimulates the maturation of oligodendrocyte precursor cells, attenuates demyelination, and enhances remyelination in the cuprizone mouse model of myelin damage and repair.

Our studies performed on cultured oligodendrocyte precursors showed that D-Asp treatment dose-dependently upregulated the transcripts and protein levels of myelin markers and this increase was significantly prevented by the NMDA receptor antagonist MK-801. More interestingly, our functional studies showed that D-Asp exposure elicited a complex $[Ca^{2+}]_i$ response in OPC involving an orchestrated functional crosstalk between glutamate transporters, ionotropic AMPA and NMDA glutamate receptors, and NCX3 exchangers. Indeed, while blockade of AMPA or NMDA receptors or NCX3 exchanger significantly prevented D-Asp-induced $[Ca^{2+}]_i$ oscillations but only partially affected the initial $[Ca^{2+}]_i$ rise, we found that blocking glutamate transporters completely prevented both the initial and oscillatory $[Ca^{2+}]_i$ response in primary OPC. In accordance with our findings, previous studies demonstrated that the sodium-dependent glutamate transporters, beyond extracellular glutamate/D-Aspartate clearance, evoked functional responses in NG2 glia (Martinez-Lozada *et al*, 2014; Moshrefi-Ravasdjani *et al*, 2018). In fact, intracellular sodium elevation upon activation of glutamate/D-Aspartate uptake has been associated with increased $[Ca^{2+}]_i$ signaling leading to a phosphorylation of the calcium/calmodulin-dependent kinase type II (CaMKII), and a promotion oligodendrocyte maturation (Martinez-Lozada *et al*, 2014). Moreover, our studies suggest that the action of D-Aspartate we observed on $Ca^{2+}$ transients in primary OPC is consequent to a cooperative activation of the sodium-dependent glutamate transporter and AMPA receptors, which then leads to secondary NMDA receptor effect. Consistently, we found that D-Asp-induced inward currents in primary OPC were unaffected by inhibition of NMDA receptors, but completely prevented by combined application of the glutamate transporter and AMPA inhibitors. In line, previous studies demonstrated that D-Asp may influence both NMDA-dependent and NMDA-independent processes (Errico *et al*, 2008a; D'Aniello *et al*, 2011). However, although the role of D-Asp as NMDA receptor agonist has been well established, it has been reported that D-Asp can either inhibit AMPA receptor-mediated currents in rat hippocampal neurons (Gong *et al*, 2005) or it may activate AMPA receptors, as it was observed in the present study in OPC and by others in dopaminergic neurons (Krashia *et al*, 2016). This observation suggests that response to D-Asp may possibly depend on the subunit composition of AMPA receptors expressed in different brain cells. Consistent with the OPC maturation promoting effects of D-Asp through a mechanism involving the activation of both AMPA and NMDA receptors, it has been shown that glutamate signaling regulates both the initiation and completion of oligodendrocyte differentiation through the concerted activation of both these ionotropic glutamate receptors. In fact, whereas the activation of AMPA receptors regulates the early phase of OPC proliferation and initial

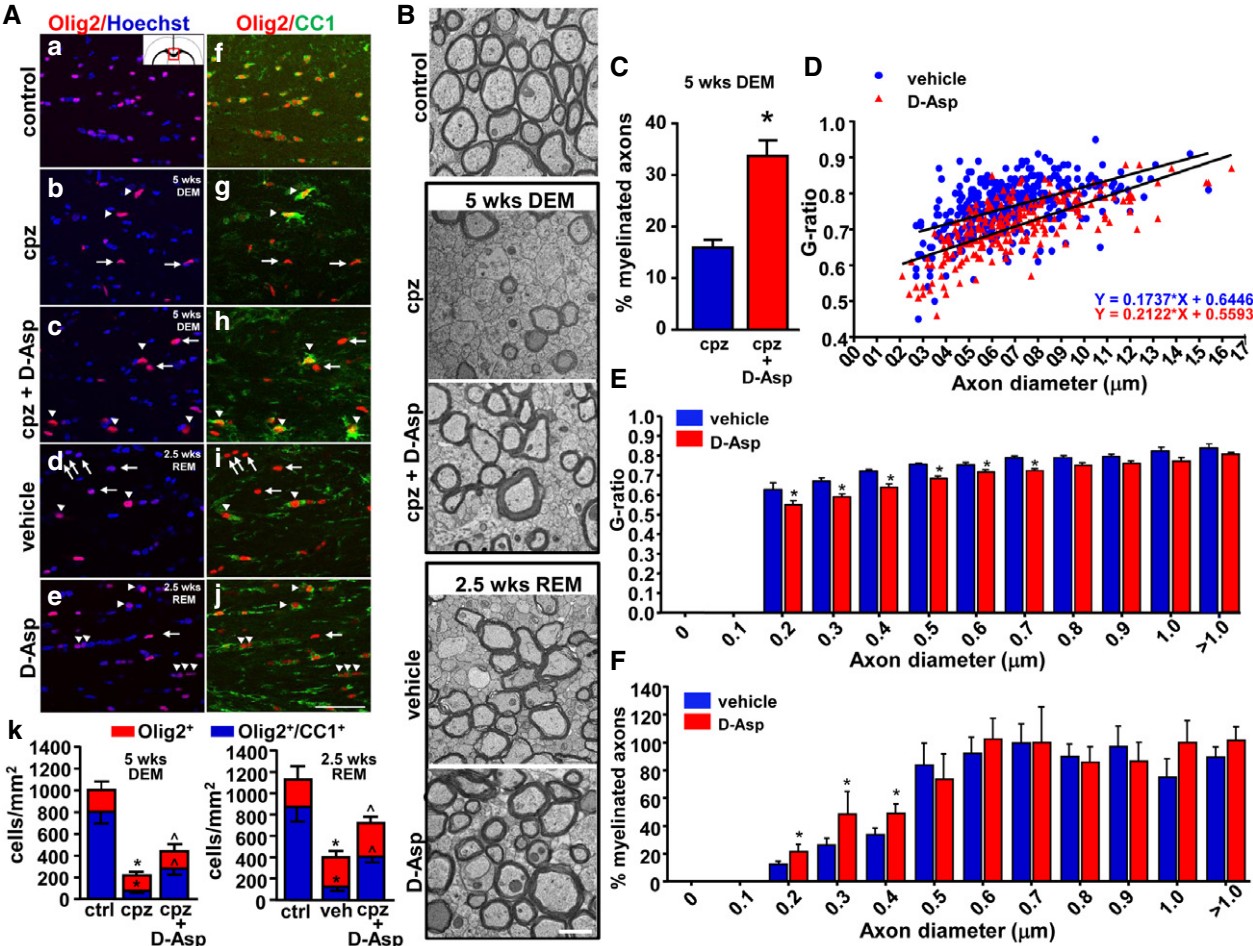

**Figure 8.    Effects of D-Aspartate treatment on *corpus callosum* remyelination.**

A    Confocal double immunofluorescence images depicting Olig2[+] (a–e) and Olig2[+]-CC1[+] (f–j) double-labeled cells in the middle *corpus callosum* of control, cuprizone-treated mice at 5 weeks, in the absence (cpz) or in the presence of D-Asp (cpz + D-Asp), and 2.5 weeks after cuprizone withdrawal, in the absence (vehicle) or in the presence of D-Asp (D-Asp). (k, left panel), Quantitative analysis of Olig2[+] cells (red) and Olig2[+]-CC1[+] (blue) double-labeled cells in the middle *corpus callosum* of control, cuprizone-treated mice, and cuprizone+D-Asp-treated mice at 5 weeks. (k, right panel), Quantitative analysis of Olig2[+] cells (red) and Olig2[+]-CC1[+] (blue) double-labeled cells in the middle *corpus callosum* of control, vehicle, and D-Asp-treated mice 2.5 weeks after cuprizone withdrawal. Arrows point to Olig2[+]-CC1[+] cells. Arrowheads point to Olig2[+]-CC1[+] cells. Scale bar in a–j: 50 μm.

B    Representative electron microscopy images of middle *corpus callosum* sections from control mice (control), from cuprizone-treated mice for 5 weeks in the absence (cpz) or in the presence of D-Asp (cpz+D-Asp), and from mice 2.5 weeks after cuprizone withdrawal in the absence (vehicle) or in the presence of D-Asp (D-Asp). Scale bars: 1 μm.

C    Percentage of myelinated axons in the middle *corpus callosum* of cuprizone (blue) and cuprizone + D-Asp-treated mice (red) for 5 weeks.

D    Scatter plot analysis of myelin thickness, expressed as individual G-ratios against axon diameters, after 2.5 weeks of cuprizone withdrawal in the middle *corpus callosum* of vehicle (blue circle) or D-Asp-treated mice (red triangle). Linear regression was used to indicate the differences between the groups in myelin thickness across the range of axon diameters.

E    Histogram analysis displaying the G-ratio values of myelinated fibers across the range of axon diameters. Data represent the mean ± SEM for at least 100 axons from each group.

F    Percentage of myelinated axons after 2.5 weeks of cuprizone withdrawal in the middle *corpus callosum* of vehicle (blue) or D-Asp-treated mice (red). DEM, demyelination; REM, remyelination.

Data information: The values represent the means ± SEM. Level of significance was determined by using: in (A, k) left panel, one-way ANOVA $P < 0.0001$ followed by Newman–Keuls *post hoc* test, *$P < 0.05$ versus ctrl; ^$P < 0.05$ versus cpz ($n = 3$ mice for each group). (A, k) right panel, one-way ANOVA $P < 0.0001$ followed by Newman–Keuls *post hoc* test, *$P < 0.05$ versus ctrl; ^$P < 0.05$ versus vehicle ($n = 3$ mice for each group); (C) two-tailed Student's *t*-test, *$P < 0.05$ versus cpz ($n = 4$ mice for each group); (E) for each axon diameter group, Student's *t*-test, *$P < 0.05$ versus vehicle ($n = 4$ mice for each group); (F) for each axon diameter group, Student's *t*-test, *$P < 0.05$ versus vehicle ($n = 4$ mice for each group). See the exact *P*-values from comparisons tests in Appendix Table S8.

Source data are available online for this figure.

differentiation, NMDA receptor activity regulates the morphological aspects of oligodendrocytes and the transition of immature to mature oligodendrocyte stage (Wake *et al*, 2011; Cavaliere *et al*, 2012; Cao & Yao, 2013; Li *et al*, 2013; Lundgaard *et al*, 2013; Martinez-Lozada *et al*, 2014). Remarkably, the pivotal role played by these receptors in orchestrating OPC development is highlighted

by recent studies showing that AMPA receptor signaling in oligodendrocyte lineage cells promotes OPC maturation and postnatal myelination by stimulating survival of newly differentiating oligodendrocytes (Kougioumtzidou *et al*, 2017), and blocking AMPA/kainate or NMDA receptors activity significantly inhibits remyelination (Li *et al*, 2013; Lundgaard *et al*, 2013; Gautier *et al*, 2015).

In line with the key role of AMPA receptors in mediating D-Asp effects on oligodendrocyte progenitors, we found that blocking AMPA receptors completely prevented both the initial and oscillatory $[Ca^{2+}]_i$ response as well as D-Aspartate-induced inward currents in human MO3.13 oligodendrocyte progenitors. The full abrogative effect of AMPA receptor blockade on D-Asp-induced $[Ca^{2+}]_i$ in MO3.13 progenitors if compared to primary OPC might be explained by several reasons. In fact, it should be taken into consideration that MO3.13 cell line is not purely human oligodendrocyte precursor cell line, but rather a hybrid line result of fusion of human rhabdomyosarcoma and adult human oligodendrocytes (McLaurin *et al*, 1995), and they are likely to replicate some but not all aspect of human oligodendrocyte precursor cell biology. In addition, our results may also suggest that undifferentiated MO3.13 progenitors, at least at very early stages, may differ from OPC for the functional expression of glutamate/D-Aspartate transporters. Although this aspect required further investigation, this hypothesis could help to explain the cycling behavior of MO3.13 cell line compared to primary OPC after D-Asp exposure. In fact, it has been demonstrated that the absence of selective glutamate transporters contributes to glutamate-induced proliferative signaling (Vanhoutte & Hermans, 2008). Whether the cell cycling effects of D-Aspartate observed in the present study may be dependent to the functional expression of glutamate transporters needs to be explored.

Another interesting finding of our study is that D-Asp exposure significantly and dose-dependently upregulated the transcripts levels of the $Na^+/Ca^{2+}$ exchanger NCX3, a plasma membrane component of myelin membranes (Gopalakrishnan *et al*, 2013) whose increased expression and activity has been shown to be essential for controlling $[Na^+]_i$ and $[Ca^{2+}]_i$ in OPC during differentiation and myelin synthesis (Boscia *et al*, 2012, 2016a; Friess *et al*, 2016). Consistent with the boosting effects of D-Asp on OPC development through a mechanisms involving the activation of both glutamate receptors and NCX3 exchangers, we found that the upregulation of NCX3 transcripts following D-Asp exposure was significantly prevented by the NMDA receptor blocker and the pharmacological blockade of NCX3 exchanger significantly prevented myelin marker increase after D-Asp exposure. Indeed, our findings propose that NCX3 operating in the reverse mode could be involved in D-Asp-induced $[Ca^{2+}]_i$ response. In line with our hypothesis, it has been demonstrated that the activation of oligodendroglial sodium-dependent glutamate transporters, by increasing $[Na^+]_i$, may participate to D-Asp-induced $[Ca^{2+}]_i$ increase (Martinez-Lozada *et al*, 2014). The relevant contribution of NCX3 to D-Asp-induced calcium response was further suggested by our results showing that knocking out *ncx3* in OPC cultures by using silencing or transgenic approaches completely prevented $[Ca^{2+}]_i$ oscillations. Indeed, agonist-evoked $[Ca^{2+}]_i$ oscillations are a characteristic property of cells expressing some receptors, including AMPA receptors, and represent a signaling system that regulates numerous processes in all cell types including proliferation and cellular differentiation (Dolmetsch *et al*, 1998).

Recently, Krasnow *et al* (2018) provide evidence that calcium transients in developing oligodendrocytes, including those evoked by neuronal activity, drive myelin sheath elongation presumably controlling proteins regulating cytoskeletal growth and myelin assembly. In this context, and in line with our findings showing the relevant contribution of NCX3 to D-Asp-evoked $[Ca^{2+}]_i$ oscillations in OPC, a very recent study demonstrated that NCX-mediated $Ca^{2+}$ influx is required for sustaining spontaneous $[Ca^{2+}]_i$ oscillations occurring in differentiating oligodendrocytes at DIV4-5 in cultures (Hammann *et al*, 2018). Based on these observations, we can speculate that D-Asp exposure, by promoting $[Ca^{2+}]_i$ oscillations, may shift the onset of spontaneous calcium activity to earlier time period, thus triggering the developmental program and accelerating oligodendrocyte differentiation.

The most interesting part of our study indicated that D-Aspartate treatment may have a beneficial effect on oligodendrocyte lineage cells in the cuprizone model during both demyelination and remyelination. In both conditions, D-Asp treatment significantly improved motor coordination and balance. In line, D-Asp treatment significantly prevented oligodendrocytes and myelin loss and partially counteracted the upregulation of astrocyte and microglial inflammatory markers in the *corpus callosum* during cuprizone-induced demyelination. In accordance with our findings, a mounting number of studies suggest the potential beneficial effects of exogenous D-Asp administration in brain pathological conditions related to myelin dysfunction and hypofunctioning of NMDA receptors, including schizophrenia and cognitive deficits (Errico *et al*, 2015; Sacchi *et al*, 2017). In fact, chronic oral administration of D-Asp that has been proved to be safe even at high doses in humans (Melville *et al*, 2015) efficiently increased the levels of this D-amino acid in the rodent brain and this upregulation has been associated with improved memory (Errico *et al*, 2008b, 2011), enhanced structural and functional synaptic plasticity (Errico *et al*, 2014), and greater connectivity in cortico-hippocampal regions (Errico *et al*, 2015). Based on these observations, it is possible to speculate that the beneficial effects of D-Asp during demyelination may also involve brain plasticity-induced myelin remodeling. In line with this hypothesis, we found that D-Asp exposure stimulated axonal myelination during developmental myelination in organotypic cerebellar slice.

In addition, it should be considered that D-Asp levels in the brain are strictly regulated by its degradating enzyme DDO (D'Aniello *et al*, 1993; Errico *et al*, 2012), and previous studies very well demonstrated that the oral administration schedule known to substantially increase the levels of this D-amino acid in the mouse brain is represented by chronic treatment with a 20 mM D-Asp solution for 1 month. (Errico *et al*, 2008a,b, 2012). This observation may help to explain why, in our *in vivo* experiments, the beneficial effects of D-Asp during remyelination were observed only with long-term D-Asp treatment (3.5 weeks), and so when D-Asp treatment was initiated 1 week before cuprizone withdrawal. Moreover, for the same reason, it is unlikely that such beneficial effects on remyelination might be ascribed to protective actions against demyelination with only 1-week D-Asp treatment. In support of this observation, we found that D-Asp treatment accelerates axonal remyelination in cerebellar explants exposed to LPC, a different model of myelin damage and repair. Remarkable, D-Asp treatment *in vivo* during remyelination significantly upregulated the number of mature oligodendrocytes in the *corpus callosum* and increased the

percentage of myelinated axons with small diameter (0.2–0.4 μm). Recent findings showed that an efficient remyelination of smaller diameter axons depends on neuronal activity more than larger diameter axons and on the activation of glutamate ionotropic receptors between demyelinated axons and OPC (Gautier *et al*, 2015). In line with this observation and beside the direct effects of D-Asp on OPC we observed in the present study, we can speculate that the stimulatory effect of D-Asp on myelination of small-diameter axons may be dependent to its ability to stimulate neuronal activity. In fact, previous studies very well demonstrated that D-Asp treatment increased neuronal activity-dependent synaptic plasticity and glutamate release (Errico *et al*, 2008b; Sacchi *et al*, 2017).

Recently, it has been proposed that the use of agonists that promote AMPA receptor currents (Gautier *et al*, 2015) or the use of sex steroid hormone therapy (Bielecki *et al*, 2016) may promote OPC differentiation and myelin repair. Based on the well-known effects of D-Asp on steroidogenesis (D'Aniello, 2007; Ota *et al*, 2012), it is possible to speculate that the beneficial effect of D-Asp observed in the present study might involve multiple mechanisms of action.

Collectively, our findings indicate that exogenous D-Asp treatment might represent a novel treatment strategy for stimulating myelin recovery after demyelination. Nevertheless, a deeper understanding of glutamate signaling in brain cells during the different stages of demyelinating disease and of D-Asp signaling on glutamate receptors with respect to subunits composition and cell types is required for designing successful D-Asp repair treatment in demyelinating diseases.

# Materials and Methods

### Animals

All animal experiments and animal handling and care were in accordance with the ARRIVE guidelines and the Guide for the Care and Use of Laboratory Animals, and the experimental protocol was approved by the Animal Care and Use Committee of "Federico II" University of Naples, Italy, and Ministry of Health, Italy (#84-85/2015-RP). Male C57BL/6 mice aged 2 months and female Wistar rats (14-day timed pregnant) were obtained from Charles River Laboratories (Italy) and maintained at a constant temperature ($22 \pm 1°C$) on a 12-h light/dark cycle (lights on at 7 AM) with food and water *ad libitum*. Male C57BL/6 mice were group-housed (3–4 per cage). The pregnant dams were allowed to deliver their pups naturally; 3 or 10 days postpartum littermates were used for OPC isolation and organotypic slices, respectively. All efforts were made to minimize animal suffering and to reduce the number of animals used.

### Cell cultures

#### The human oligodendrocyte MO3.13 cell line and D-Asp exposure
The MO3.13 cell line is an immortalized human clonal model that expresses the phenotypic characteristics of OPC (McLaurin *et al*, 1995; Boscia *et al*, 2012). Cells were maintained in Dulbecco's modified Eagle's medium (DMEM) supplemented with 10% fetal bovine serum (FBS), 100 U/ml penicillin, 10 μg/ml streptomycin, and 2 mmol/l glutamine (normal medium). To induce a mature oligodendrocyte phenotype, human MO3.13 cells were differentiated in a serum-free chemically defined medium composed of DMEM supplemented with 500 μg/l insulin, 100 μg/ml human transferrin, 0.52 μg/l sodium selenite, 0.63 μg/ml progesterone, 16.2 μg/ml putrescine, 100 U/ml penicillin, 100 μg/ml streptomycin, 2 mM glutamine, 5 mg/ml N-acetyl-L-cysteine, and 10 μM D-biotin (OPC medium), in the presence of 100 nM phorbol-12-myristate-13-acetate (PMA) for 3–4 days. PMA is a natural analog of the potent protein kinase C (PKC) activator diacylglycerol, and it is traditionally utilized as a tool to activate the PKC signaling pathway. Studies demonstrated that chronic PMA treatment upregulated the expression of MBP transcripts and protein in human oligodendrocyte MO3.13 cells (McLaurin *et al*, 1995), stimulates process extension in primary oligodendrocytes (Yong *et al*, 1991), and promotes their differentiation through a mechanism involving the activation of NADPH oxidase and ROS generation (Cavaliere *et al*, 2013). Fresh 10–200 μM D-Asp (Sigma-Aldrich, Milan, Italy) was added to the cultures in the absence of PMA, by replacing the medium every day. Control cultures were kept in serum-free medium in the absence of D-Asp.

#### Primary rat and mouse OPC cultures and D-Asp exposure
Purified OPC cultures were prepared as previously described (Boscia *et al*, 2012). In brief, primary mixed glial cell cultures were obtained from the cerebral cortex of postnatal days 1–3 rats or mice pups, dissociated into single cells, and cultured into poly-D-lysine (Sigma-Aldrich, St. Louis, MO, USA)-coated tissue culture flasks in normal medium at 37°C in a humidified, 5% $CO_2$ incubator. Once confluent (after 7–9 days), the microglia were separated by mechanical shaking of flasks on a rotary shaker for 60 min at 200 rpm and removed. The cultures were then subjected to an additional 16 h of shaking at 200 rpm. To minimize contamination by microglial cells, the suspension of detached cells was incubated for 1 h at room temperature (RT). The non-adhering OPC was plated into poly-D-lysine-coated plates in normal medium and maintained at 37°C in a humidified, 5% $CO_2$ incubator. This procedure yields 98% A2B5-positive cells. Six hours after plating, the culture medium was replaced with OPC medium supplemented with 10 ng/ml recombinant human platelet-derived growth factor-AA (PDGF-AA) and 10 ng/ml basic fibroblast growth factor (bFGF) each day for 2–3 days to maintain the undifferentiated state and support OPC survival. Then, to facilitate the differentiation of precursors into oligodendrocytes, PDGF-AA and bFGF were withdrawn from the OPC medium and were instead supplemented with 100 μM D-Asp each day for 3–4 days. Control cultures were kept in OPC medium.

#### Organotypic cerebellar slice cultures and D-Asp exposure

Organotypic slice cultures were prepared as previously described (Boscia *et al*, 2006, 2013b; Zhang *et al*, 2011). Briefly, 400-μm-thick parasagittal slices were obtained from cerebellum of P8- to P10-day-old Wistar rat pups (Charles River Laboratories, Calco, Italy) using a McIlwain tissue chopper (Campden Instruments, Leicester, UK) and placed into ice-cold Hank's balanced salt solution (HBSS, Gibco-BRL, Renfrewshire, UK) supplemented with 6.5 mg/ml glucose and 1.5% (v/v) Fungizone. Cultures were then transferred to a humidified semiporous membrane (30-mm Millicell tissue culture plate inserts of 0.4 mm pore size from Millipore,

Rome, Italy) in six-well tissue culture plates (five slices per membrane). Each well contained 1.2 ml of tissue culture medium consisting of 50% minimal essential medium (MEM, Gibco, Thermo Fisher Scientific, Monza, Italy), 25% HBSS, 25% heat-inactivated horse serum (Gibco), 6.5 mg/ml glucose, 1 mM gluta-mine (Gibco), and 1.5% Fungizone (Gibco). Cultures were maintained at a 37°C and 5% $CO_2$-conditioned atmosphere. For myelination studies, slices cultures were kept in culture for 2 days in vitro (DIV) to allow tissue recovery from the dissection and then cultured in the presence or in the absence of 100 μM D-Asp from 3 DIV until 7 DIV (Fig 2A). Fresh D-Asp was replaced every day and all slices were maintained for 7 DIV, when they were collected for further analysis. For remyelination studies, at 11 days in vitro cere-bellar organotypic slices were demyelinated with 0.5 mg/ml lysophosphatidylcholine (LPC, Sigma) for 15–17 h. Then, slices were washed in normal medium for 5 min and treated at 2 days postlysolecithin (dpl) with 100 μM D-Asp until 6, and 10 dpl or vehicle controls (Fig 2E).

### Staining and image analysis of organotypic slices

Slices were fixed in 4% wt/vol paraformaldehyde in phosphate buffer for 1 h, blocked with 3% bovine serum albumin (BSA) and 0.05% Triton X (Sigma, Milan, Italy) for 40 min, and then incubated with anti-myelin basic protein (monoclonal anti-MBP; 1:1,000, #SMI-99P, Covance) and axonal neurofilament-200 (rabbit poly-clonal anti-NF200; 1:500, #N4142, Sigma) primary antibodies for 48 h at 4°C. Then, slices were incubated with the corresponding secondary antibodies (goat anti-mouse and goat anti-rabbit; each at 1:200; Thermo Fisher Scientific, Inc.) overnight at 4°C. Slices were mounted and imaged with Zeiss LSM 700 laser (Carl Zeiss) scanning confocal microscope.

Confocal microscopy was used to obtain stacks of photo-graphs of MBP and NF200 immunolabeling at 2-μm intervals in white matter areas at ×40 magnification and a resolution of 1,024 × 1,024, between a depth of 5–20 μm from the upper surface. Four to five independent slices from four different animals were analyzed per condition. Five images of randomly chosen areas of each slice were acquired with identical fluores-cence intensity.

For analysis, maximum intensity projection (MIP) images were created for each stack by using ZEN imaging software. Then, for each MIP image the myelination index was calculated by dividing the area covered by overlapping pixels (colocalized points) between MBP and NF200 immunostaining and the area covered by NF200 pixels (Boscia et al, 2017).

### Cell cycle assay and cell growth analysis

For cell cycle analysis, the cells were dissociated to single-cell suspension, fixed with cold 70% ethanol before propidium iodide (PI) staining (20 μg/ml), and analyzed by flow cytometry using a BD FACSCanto II™ cytofluorimeter (BD Biosciences). Cell cycle analysis in primary cultures was performed on OPC obtained either from cerebral cortex or from whole rat brain. Both OPC cultures give equivalent results. To determine cell growth kinetics, cells were detached using trypsin-EDTA (Gibco-BRL) and then counted in triplicate; cell counts were done twice.

### Quantitative real-time RT–PCR

Total RNA was extracted from human MO3.13 cells using TRIzol (Invitrogen, Milan, Italy), as previously described (Boscia et al, 2012). After DNase-I treatment (1 U/ml; Sigma-Aldrich) for 15 min at RT, the first-strand cDNA was synthesized using 5 mg of the total RNA and 500 ng of random primers by using the SuperScript (High-Capacity cDNA RT Kit; Applied Biosystems, Monza, Italy). Using 1/10 of the cDNAs as a template, quantitative real-time PCR was performed in a 7500 Fast Real-Time PCR System (Applied Biosystems) by using the Fast SYBR Green Master Mix (Applied Biosystems). Samples were amplified simultaneously in triplicate in one assay run for 40 cycles with a single fluores-cence measurement. PCR data were then collected by using the ABI Prism 7000 SDS software (Applied Biosystems). Afterward, the products were electrophoretically separated on 3% agarose gels and bands were visualized with ethidium bromide and docu-mented by using the Gel Doc Imaging System (Bio-Rad, Hercules, CA, USA). Normalization of data was performed by using the ribo-somal protein L19 as an internal control; changes in mRNAs were represented as the mean of the relative quantification mRNA values calculated as the difference in threshold cycle (DCt) between the target gene and the reference gene ($2^{-\Delta\Delta C_T}$). The oligonucleotide sequences were as follows: h-MBP forward: 5′-GGAAACCACG CAGGCAAACGAGA-3′ and h-MBP reverse: 5′-GAAAAGAGGCGGAT CAAGTGGGG-3′; h-CNPase forward: 5′-GGCCAC GCTGCTAGAGTG CAAGAC-3′ and h-CNPase reverse: 5′-GGTACTGGTACTGGTCGGC CATTT-3′; h-NCX1 forward: 5′-GAATGAAAGGTGGCTTCACAATAA-3′ and h-NCX1 reverse: 5′-GCCTCTCCTCTTCCTCTTTGC-3′; h-NCX3 forward: 5′-GGCTGCACCATTGGTCTCA-3′ and h-NCX3 reverse: 5′-TGCCAAATGCCACGAAAA-3′; h-L19 forward: 5′-CTAGTGTCCTCCG CTGTGG-3′ and h-L19 reverse: 5′-GATGTCACGCACGATTT-3′.

### Western blotting

Protein samples from corpus callosum were separated on 8 or 14% polyacrylamide gel and electrophoretically transferred onto nitrocel-lulose membranes, as described (Boscia et al, 2017). Filters were probed with the indicated primary antibodies: rat monoclonal anti-MBP (1:200; #MAB386, Millipore), polyclonal anti-glial fibrillary acid protein anti-GFAP (1:1,000; Clone GA5, Sigma, Milan, Italy), rabbit polyclonal anti-ionized calcium-binding adapter molecule anti-Iba1 (1:1,000; #019-19741, Wako), and anti-α-tubulin (1:1,000; #T5168, Sigma, Milan, Italy). Proteins were visualized with the corresponding peroxidase-conjugated secondary antibodies, using the enhanced chemiluminescence system (Amersham Pharmacia Biosciences LTD, Uppsala, Sweden).

### In vitro NCX3 silencing

Silencing of NCX3 in MO3.13 cells was performed by using the HiPerFect Transfection Kit (Qiagen, Milan, Italy), by using the following FlexiTube siRNAs for NCX3, Hs_Slc8A3(#8) 5′-CACCACGCTCTTGCTTCCTAA-3′, and a validated irrelevant AllS-tars siRNA as a negative control (siCtl), as previously described (Boscia et al, 2012). Cells were incubated with Opti-MEM (Invit-rogen) supplemented with the RNAiFect Transfection Reagent (Qiagen) and 20 nM of the siRNA duplex for 6 h. Cells were

incubated in OPC medium for 48 h before microfluorimetric $[Ca^{2+}]_i$ measurements.

## Microfluorimetric $[Ca^{2+}]_i$ measurement

Intracellular changes in $[Ca^{2+}]_i$ were measured by single-cell FURA-2AM video-imaging technique, as previously described (Secondo *et al*, 2006; Pannaccione *et al*, 2007; Boscia *et al*, 2009). Oligodendrocyte MO3.13 progenitors and rat primary OPC were plated on 10 μg/ml poly-D-lysine-coated glass coverslip, and after 24 h of incubation in OPC medium were loaded with 6 μM FURA-2 AM for 30 min at 37°C in normal Krebs solution containing 5.5 mM KCl, 160 mM NaCl, 1.2 mM $MgCl_2$, 1.5 mM $CaCl_2$, 10 mM glucose, and 10 mM HEPES-NaOH (pH 7.4). Then, coverslips were placed into a perfusion chamber (Medical System Co., Greenvale, NY, USA) and mounted onto the stage of an inverted Zeiss Axiovert 200 Microscope (Carl Zeiss, Milan, Italy). Images were acquired with a FLUAR ×40 oil objective, whereas cells were alternatively illuminated at wavelengths of 340 and 380 nm by a xenon lamp.

$[Ca^{2+}]_i$ oscillations were identified and quantified using a computer program written in Java computer language as reported (Secondo *et al*, 2006). Briefly, for each single cell, the software calculated the $[Ca^{2+}]_i$ mean ± SD during the baseline recording interval before drug exposure; these values were used to define a cutoff to identify $[Ca^{2+}]_i$ oscillation, which was set at mean $[Ca^{2+}]_i$ ± 1 SD. Subsequently, the software identified as a single $[Ca^{2+}]_i$ oscillation each $[Ca^{2+}]_i$ value higher than this cutoff point. To quantify the effect of specific pharmacological treatments on the occurrence of $[Ca^{2+}]_i$ oscillations, the oscillation frequency was used to define the number of peaks divided by the duration of observation. In control experiments, no significant changes in the frequency occurred after the addition of drug vehicle.

The initial calcium peak was quantified and expressed as Δ% of $[Ca^{2+}]_i$ peak versus basal values of calcium. The selective and competitive AMPA receptor antagonist, 6,7 dinitroquinoxaline-2,3-dione (DNQX), was preincubated with AMPA for 10 min to block the channel in a use-dependent way before AMPA stimulation.

## Electrophysiology

AMPA currents were recorded from human oligodendrocyte MO3.13 precursors and rat primary OPC by using patch-clamp technique in whole-cell configuration using a commercially available amplifier Axopatch 200B (Molecular Devices, CA, USA), and data were acquired with a Digidata 1322A acquisition system (Molecular Devices) and pCLAMP 10 software (Molecular Devices; Pannaccione *et al*, 2007; Boscia *et al*, 2017). Patch borosilicate glass pipettes were prepared with a puller (Narishige, PC-10, Tokyo, Japan). The resistance of the pipette was 4–5 MΩ. The dialyzing pipette solution contained the following (in mM): 100 Cs-gluconate, 10 TEA, 20 NaCl, 1 Mg-ATP, 0.1 $CaCl_2$, 2 $MgCl_2$, 0.75 EGTA, and 10 HEPES, adjusted to pH 7.2 with CsOH. The cells were perfused with external Ringer's solution containing the following (in mM): 126 NaCl, 1.2 $NaHPO_4$, 2.4 KCl, 2.4 $CaCl_2$, 1.2 $MgCl_2$, 10 glucose, and 18 $NaHCO_3$, pH 7.4. The holding potential was maintained at −70 mV in order to record AMPA currents (Sekiguchi *et al*, 2002). Signals were low-pass filtered at 5 kHz and sampled at 10 kHz. Drugs were applied

using a handheld pipette and used at the following concentrations: 1–100 μM AMPA, 100 μM D-Aspartate, and 10 μM DNQX, 10 μM MK-801, and 20 μM L-trans-pyrrolidine-2,4-dicarboxylic acid (PDC).

## Cuprizone-induced demyelination/remyelination and D-Asp treatment

Experimental toxic demyelination was induced by feeding 8-week-old male C57BL/6 mice a diet with 0.2% (w/w) cuprizone (oxalic bis-cyclohexylidenehydrazide; Sigma-Aldrich, Milan, Italy) mixed into milled chow pellets (Harlan Laboratories, Milan, Italy). Food containing cuprizone was changed every 2 days for 5 weeks. D-Asp (Sigma-Aldrich, Milan, Italy) was delivered in drinking water at the concentration of 20 mM according to previous *in vivo* studies (Errico *et al*, 2008a). The effects of D-Asp treatment were investigated during cuprizone-induced demyelination and during remyelination, after cuprizone withdrawal (Fig 7A). Water or D-Asp drinking solution was available *ad libitum* for all groups of mice, consuming from 4 to 6 ml each in 24 h.

To analyze the effects of D-Asp on acute demyelination (DEM), animals were divided into three groups: (i) control mice, which were fed with normal chow for all the time of experiments (5 weeks), (ii) cuprizone mice, which were fed with 0.2% cuprizone for 5 weeks, and (iii) cuprizone plus D-Asp mice, which were fed with 0.2% cuprizone and received 20 mM D-Asp in drinking solution for 5 weeks (Fig 7A).

To analyze the effects of D-Asp on remyelination (REM), animals were divided into four groups: (i) control mice, which were fed with normal chow for all the time of experiments (7.5 weeks), (ii) vehicle mice, which were fed with 0.2% cuprizone for 5 weeks, and maintained with normal chow for additional 2.5 weeks after cuprizone withdrawal, (iii) D-Asp (I) mice, which received cuprizone for 5 weeks, and 20 mM D-Asp in drinking solution for 2.5 weeks after cuprizone withdrawal, and (iv) D-Asp (II) mice, which received cuprizone for 5 weeks, and 20 mM D-Asp in drinking solution starting 1 week before cuprizone withdrawal and for 2.5 weeks after cuprizone treatment (Fig 7A). Six to eight animals per group were used.

## Behavioral testing

Motor coordination and balance were evaluated by using the beam balance and rotarod tests. The beam apparatus consisted in 1 cm wide and 50 cm long beam, elevated 50 cm from the work surface with a 10-degree inclination. In order to prevent mice injury, a soft sawdust pad attenuates the mice foot-slips. During the training session, mice were individually placed at the start of the beam and allowed to freely transverse till the end of the runway where a resting box was located. After training, mice returned to their home cages. The test consisted of three trials spaced out at least 5 min. The experiment was repeated for the 3 consecutive days. During the session test, the latency to transverse the beam was measured.

For rotarod tests, mice were placed on a rotating cylindrical rod (Ugo Basile, Varese, Italy), and both fixed-speed and accelerating rotarod protocols were applied. In the fixed-speed protocol, the apparatus rotation was set at a constant rate of 16 revolutions per minute (rpm) and the number of falls was recorded during 60 s. In

the accelerating rotarod protocol, the rod was accelerated (20 rpm/min, from 0 to 16 rpm in 48 s) and then maintained at constant rate of 16 rpm rotation. The latency to fall from the rod was automatically recorded for each animal. When the animal safely drops into a plastic lane below, the test is considered concluded and the final time recorded (maximum 3 min). The experiment consisted in three consecutive sessions conducted with an inter-trial interval of 10 min. The experiment was repeated for 3 consecutive days. The mean number of falls and latency to fall is the datum for three trials per day executed for 3 consecutive days.

Mice completed three trials for each task during 1, 3, and 5 weeks of cuprizone treatment (demyelination) and 2 weeks after cuprizone withdrawal (remyelination).

### Confocal immunofluorescence analysis

Confocal immunofluorescence procedures in cells or sections were performed as described previously (Boscia *et al*, 2008, 2016b). Mice were deeply anesthetized with Zoletil 100 (zolazepam/tile-tamine, 1:1, 10 mg/kg, Laboratoire Virbac) and Xilor (xylazine 2%, 0.06 ml/kg, Bio98), and transcardially perfused with 4% (wt/vol) paraformaldehyde in phosphate buffer. Brains were cryoprotected in sucrose, frozen, and sectioned coronally at 50 μm on a cryostat. Cell cultures were fixed in 4% wt/vol. paraformaldehyde in phosphate buffer for 30 min. After blocking with Rodent M Block (Biocare Medical, Concord, CA, USA) or BSA 3%, sections or cells were incubated with primary antibodies for 24 or 48 h, respectively. The primary antibodies used were the following: rabbit polyclonal anti-NF200 (1:1,000, #N4142, Sigma, Milan, Italy); rat monoclonal anti-MBP (1:400, #MAB386, Millipore); mouse monoclonal anti-myelin-associated glycoprotein (MAG, 1:500, #MAB1567, Millipore); mouse monoclonal anti-glial fibrillary acidic protein (1:1,000; Clone GA5, Sigma, Milan, Italy); rabbit polyclonal anti-ionized calcium-binding adapter molecule 1 (1:2,000, Iba1, #019-19741, Wako Pure Chemical Industries, Ltd., Osaka, Japan); rabbit polyclonal anti-NG2 chondroitin sulfate proteoglycan (1:400, #AB5320, Millipore); rabbit polyclonal anti-Olig2 (1:200, #AB9610, Millipore); and mouse monoclonal anti-adenomatous polyposis coli (anti-CC1, 1:2,000, #ab16794, Abcam). Then, cells or sections were incubated with corresponding fluorescence-labeled secondary antibodies (Alexa488- or Alexa594-conjugated anti-mouse or anti-rabbit IgGs). Hoechst-33258 was used to stain nuclei. Images were observed using a Zeiss LSM 700 laser (Carl Zeiss) scanning confocal microscope. Single images were taken with an optical thickness of 0.7 μm and a resolution of 1,024 × 1,024. All staining and morphological analyses were blindly conducted.

Oligodendroglial cell morphology *in vitro* was evaluated with Alexa594-phalloidin (Thermo Fisher Scientific, Milan, Italy) and with MAG or MBP antibodies after 4 days both in D-Asp-treated and in D-Asp-untreated cultures as described by Sperber and McMorris (2001) and Zuchero *et al* (2015), with slight modifications. Individual MAG-phalloidin-positive or MBP-phalloidin-positive oligodendrocytes were scored at 4 days according to their morphological complexity in four categories: (i) *Low*, including cells with bipolar or tripolar morphology, (ii) *Medium*, including multipolar cells with simple arborization complexity, (iii) *Medium–High*, including cells

**The paper explained**

**Problem**
Promoting myelin repair holds great potential to improve the lives of multiple sclerosis (MS) patients. Novel pharmacological compounds able to boost oligodendrocyte maturation and axonal remyelination need to be identified.
In recent years, glutamatergic signaling at axo-myelinic synapses was suggested to regulate myelin remodeling and repair. D-Aspartate is a D-amino acid exerting modulatory actions at glutamatergic synapses. Whether exogenous D-Aspartate treatment could represent a novel strategy to promote myelin repair is thus far unknown.

**Results**
We here report the role of D-Aspartate signaling in myelination during healthy development and following injury. Using *in vitro* models of oligodendrocyte differentiation and myelination, we demonstrated that D-Aspartate exposure stimulated progenitor differentiation into myelin-producing oligodendrocytes and accelerated developmental myelination in cerebellar organotypic slices. Using *in vivo* models of myelin damage and repair, we show that D-Aspartate treatment may have beneficial effects on oligodendrocyte lineage cells during both demyelination and remyelination. Indeed, administration of D-Aspartate improved motor coordination, accelerated myelin recovery, and significantly increased the number of small-diameter myelinated axons. Our functional studies demonstrated that D-Aspartate exposure elicited a complex $[Ca^{2+}]_i$ response in oligodendrocyte precursors involving an orchestrated functional crosstalk between glutamate transporters, ionotropic AMPA and NMDA glutamate receptors, and the $Na^+/Ca^{2+}$ exchanger NCX3.

**Impact**
Results provided in this manuscript indicate that exogenous D-Aspartate treatment offers significant promise as treatment strategy for attenuating myelin damage and stimulating myelin recovery following a demyelinating injury.

with complex radially oriented arborization pattern, and (iv) *High*, including cells displaying compact MBP-positive regions (MBP$^{++}$). Cell counts of MAG$^+$ and MBP$^+$ cells in oligodendrocyte cultures were performed manually. MBP$^{++}$ cells refer only to cells displaying clear compact MBP-positive regions. Oligodendroglial cell morphology in pharmacological studies was evaluated with NG2 and MAG antibodies after 3 days of incubation. Cell counts of NG2$^+$, MAG$^+$, and NG2$^+$-MAG$^+$ double-labeled cells in oligodendrocyte cultures were performed manually.

Quantification of MBP and NF200 fluorescence intensity within the middle *corpus callosum* on tissue sections at the level of the septostriatal region was quantified in terms of pixel intensity value by using the NIH image software, as described previously (Boscia *et al*, 2017). Briefly, digital images were taken with 63× objective and identical laser power settings and exposure times were applied to all the photographs from each experimental set. Images from the same areas of the *corpus callosum* were compared. The quantification of colocalization between MBP and NF200 immunostaining was assessed by using the "colocalization highlighter" plug-in for ImageJ Software (NIH, Bethesda, MA, USA). Before colocalization analysis, threshold settings for each image were determined, and quantification was achieved by counting the number of MBP-NF200 colocalized points (white) per microscope field. Results were expressed as a percentage of colocalization. Six sections from each

mouse were analyzed, with $n$ = 3–4 mice per treatment group, for a total of 18 sections per treatment group.

The number of GFAP$^+$, Iba1$^+$, Olig2$^+$ (representing all oligodendrocyte lineage cells), and Olig2$^+$/CC1$^+$ cells (representing mature oligodendrocytes) was determined in the middle *corpus callosum* by manual counting at ×40 magnification. Only cells with clearly a visible cell body and profiles were counted. Four mice per group were included in the studies, and six slices from every mouse were analyzed.

### Electron microscopy

Mice were deeply anesthetized with Zoletil 100 (zolazepam/tiletamine, 1:1, 10 mg/kg, Laboratoire Virbac) and Xilor (xylazine 2%, 0.06 ml/kg, Bio98), and transcardially perfused with 4% (wt/vol) paraformaldehyde and 2.5% (wt/vol) glutaraldehyde in phosphate buffer. *Corpus callosum* sections were sectioned coronally on a vibratome and postfixed in 1% uranyl acetate and 1% OsO4. After dehydration through a graded series of ethanol, the tissue samples were cleared in propylene oxide, embedded in the Epoxy resin (Epon 812), and polymerized at 60°C for 72 h. From each sample, ultrathin sections were cut on a Leica EM UC7 Ultramicrotome. Thin sections were further investigated using a FEI Tecnai-12 (FEI, Eindhoven, The Netherlands) electron microscope equipped with a Veletta CCD camera for digital image acquisition. Brain sections within the medial *corpus callosum* at the level of septostriatal region were obtained. Sections were evaluated from each mouse (three mice per group), and at least 5–6 fields per mouse were taken. The number of myelinated axons and their G-ratios (the ratio of axon diameter to the outside diameter of the myelin) were analyzed using ImageJ software by blinded analysis. Non-serial sections were evaluated, and 100 axons per mouse were counted.

### Statistical analysis

Sample sizes were chosen based on previous experience and based upon similar studies from the literature. In *in vitro* studies, at least three independent replicates were conducted to ensure reproducibility. In animal studies, we used $n$ = 7–14 mice/group for behavioral tests and $n$ = 4–5 mice/group for biochemical, confocal, and electron microscopy analyses. No animals were excluded from the analysis. Sample size and number of replicates are indicated in figure legends. All mice were age- and weight-matched. No additional randomization was used. For all experiments, the animals were assigned to different group by random selection. For *in vivo* pharmacological treatments, the investigator was not blinded to the experimental group. Sample processing, quantification in microscopy studies, and recording the latency were performed in a blinded manner. The data are expressed as the mean ± SEM of the values obtained from individual experiments. Statistical comparisons between groups were performed by Student's $t$-test or one-way analysis of variance (ANOVA) followed by Bonferroni, Tukey, or Newman–Keuls *post hoc* test; $n$ indicates the number of experiments. A difference of $P < 0.05$ was considered significant. GraphPad Prism 6.0 was used for statistical analysis (GraphPad Software, Inc, La Jolla, CA, USA).

**Expanded View** for this article is available online.

## Acknowledgements

This work was supported by grants from **Fondazione Italiana Sclerosi Multipla FISM 2015/R/06** to F.B.; Programma Operativo Nazionale (ARS01_01226 and PON03PE_00146_1) from MIUR to L.A. We thanks Dr S. Sokolow (UCLA School of Nursing, Los Angeles, CA, USA) and Dr A. Herchuelz (Laboratory of Pharmacology and Therapeutics, Universite' Libre de Bruxelles, Brussels, 6041 Gosselies, Belgium) for providing the ncx3$^{-/-}$ mice, and Dr Laura Pisapia and Pasquale Barba for technical assistance, FACS Facility of IGB-CNR, Naples. The authors thank Prof. Alessandro Usiello (Universita' degli Studi della Campania "Luigi Vanvitelli", Napoli, Italy) for helpful discussion.

## Author contributions

FB conceived the project and designed research; VdR, AS, AP, RCi, LF, NG, AF, and FB performed research and analyzed data; RP and RCr performed electron microscopy experiments; VdR, AS, AP, LA, and FB edited figures and provided advice; AD'A, LA, and FB contributed to reagents/analytic tools; and FB written the manuscript.

## Conflict of interest

The authors declare that they have no conflict of interest.

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
