## [Review Process File · EMBO Molecular Medicine]

D-Aspartate treatment attenuates myelin damage and stimulates myelin repair

Valeria de Rosa, Agnese Secondo, Anna Pannaccione, Roselia Ciccone, Luigi Formisano, Natascia Guida, Roberta Crispino, Annalisa Fico, Roman Polishchuk, Antimo D'Aniello, Lucio Annunziato, Francesca Boscia

Review timeline:	Submission date:	28 April 2018
	Editorial Decision:	28 May 2018
	Revision received:	8 October 2018
	Editorial Decision:	6 November 2018
	Revision received:	20 November 2018
	Accepted:	22 November 2018

Editor: Céline Carret

Transaction Report:

1st Editorial Decision

28 May 2018

Thank you for the submission of your manuscript to EMBO Molecular Medicine. We have now heard back from the two referees whom we asked to evaluate your manuscript.

You will see from the comments pasted below, that both reviewers are overall supportive of publication. This said, additional experiments in a different model system (ref.1), with different time settings (ref.1 and 2), verify main results in primary cells (ref.2) are requested, which if performed, would strengthen the data considerably. Referees also would need some clarifications and more discussion of the results, source data and better statistics and display are also expected.

We would therefore welcome the submission of a revised version within three months for further consideration and would like to encourage you to address all the criticisms raised as suggested to improve conclusiveness and clarity. Please note that EMBO Molecular Medicine strongly supports a single round of revision and that, as acceptance or rejection of the manuscript will depend on another round of review, your responses should be as complete as possible.

I look forward to receiving your revised manuscript.

***** Reviewer's comments *****

Referee #1 (Comments on Novelty/Model System for Author):

The authors should consider using an alternative model of demyelination and remyelination, to that of the cuprizone model, to test the effects of d-aspartate on myelin damage and repair. To verify that the effect is not specific to cuprizone mediated damage.

Referee #1 (Remarks for Author):

In this study de Rosa and colleagues investigate the role of d-aspartate signaling in regulating OPC fate and subsequently the role of d-aspartate in demyelination and remyelination. The authors used an impressive array of approaches, from human to rodent *in vitro* cultures, organotypic cell cultures, electrophysiology, calcium imaging, behavioural and *in vivo* models of myelin damage and repair to identify the role of d-aspartate on OPCs fate, its mechanisms of action and its potential as a therapeutic agent for myelin disorders. Overall this is an important study in a very topical area.

The main findings were that d-aspartate treatment, *in vitro*, enhances OPC cell cycle time, myelin gene expression, and differentiation into myelinating oligodendrocytes. They identify d-aspartate evokes intracellular calcium rise via activation of NMDA and AMPA receptors, and subsequently via the Na/Ca exchanger. Lastly, they show that d-aspartate rescues the loss of motor function and myelin damage in the cuprizone model, a model of demyelination. As d-aspartate can be safely used in humans, this study has important therapeutic potential. However, the fact that the therapeutic effect was only observed when d-aspartate was given in the presence of cuprizone questions its mode of action.

Comments/questions

- The finding that d-aspartate does is not beneficial after cuprizone treatment, but only when given in the presence of the copper chelator cuprizone indicates that d-aspartate is protective against cuprizone-mediated OL death and demyelination than in improving OPC differentiation and remyelination. It is difficult to comprehend the lack of an effect on remyelination, given the potentiating effect that d-aspartate has on OPC differentiation *in vitro*. Therefore to substantiate whether d-aspartate is only protective against cuprizone mediated toxicity (not unlikely given the complicated effects of copper on glutamate receptor signaling and in particular NMDA receptors) or has a generic protective role, the authors should try to use another animal model of myelin repair, such as lysolecithin LPC or equivalent model which mode of damage does not interfere with glutamate or nrx signalling. The authors should also verify whether remyelination would be evident if the animals were kept longer post-cuprizone treatment thus allowing for an investigation into the role of d-aspartate on remyelination and on the cellular mechanism of repair, and thus separating the role of d-aspartate on damage and repair?

- The effects of d-aspartate on AMPA and NMDA receptors are intriguing. However, it is not clear whether the effect of MK801 on the first intracellular calcium peak is due to the fact that MK801 is an open channel blocker and thus may not block the d-aspartate-evoked calcium until after it is applied, or whether the action of d-aspartate is via AMPA receptors, which then leads to secondary NMDA receptor effect. As result further questions arise, does d-aspartate evoke NMDA receptor mediate current in OPCs? If so would APV, a competitive agonist inhibit the calcium signal in the same way as DNQX? Then can the author explain the mode of action, if OPCs express both NMDA and AMPA receptors how come the d- aspartate-evoked current is completely blocked by AMPA antagonist?

Minor comments:

(1) Explanation of acronym should be given in text and not only in methods, for example, PMA. Similarly, information that it activates PKC signaling, and the rationale for using PMA should also be provided in the text.

(2) Quantification of western blots should be provided in all figures where the blots are shown, similarly to Figure 7B

(3) page 5 second line - first paragraph, '...human MO3.13 precursor or primary OPCs...' rodent or rat should be inserted in front of OPCs in this sentence.

(4) Page 4, 11th line from top, first paragraph states '...copper chelator cuprizone an in vivo model of MS...' - cuprizone model is not a model of MS. This sentence needs to be rewritten to highlight that it is a model of myelin damage and repair, an important aspect in terms of understanding the regenerative process of MS, but not a model of MS.

Referee #2 (Comments on Novelty/Model System for Author):

The authors presenting a series of experiments that conclusively suggest that D-aspartate as potential to be used for remyelination strategies in a pathological context.

Referee #2 (Remarks for Author):

In the manuscript "D-Aspartate treatment attenuates myelin damage and stimulates myelin repair", de Rosa and colleagues investigate the role of D-aspartate in oligodendrocyte lineage progression and present data that this amino acid can have different roles in this progression and can lead to less demyelination and more remyelination in a cuprizone mouse model. The data is solid, with the authors presenting a series of experiments that conclusively suggest that D-aspartate has potential to be used for remyelination strategies in a pathological context. Therefore, I recommend publication in EMBO Molecular Medicine, with minor revisions addressing the following points:

1) M03.13 cell line is not purely human oligodendrocyte precursor cell line, but rather a hybrid line result of fusion of human rhabdomyosarcoma and adult human oligodendrocytes. The authors should clearly state this in the manuscript since these cells are likely to replicate some but not all aspect of human oligodendrocyte precursor cell biology. While the authors do replicate most findings also in rat primary OPCs, the findings of Figure 1c (effects of D-aspartate in cell cycle) and Figure 5 (effects of AMPA) should be replicated in primary rat OPCs.

2) The effects of D-aspartate in cell cycle (Fig. 1c) appear to be transient (only at day 3) and I am not convinced that they are relevant. Why is the effect only observed after 3 days? The authors observe a stabilization of cell numbers with D-Asp treatment at day 4, but what happens subsequently? Is the number of cells in control also stabilized at day 5 and beyond, or do they continue to increase while they remain stable with D-asp treatment? Addition of these time points would help clarify the role of D-asp in proliferation. Also, according to the figure legend, n=2 in these assays, which should be increased.

3) In the western blots presented, the control lanes are separated from the D-Asp treatment. I assume these western blots are from the same gel but with intermediate bands missing, the authors should present the whole blot in Supplementary material. It is also not clear from the figure legends how many replicates the presented western blots are representative of. In addition, in Fig. 1C, the band corresponding to 100uM is narrower than the other bands. I would advice the authors to present in the main figure another western blot where all the lanes have the same width, and the remaining western blots as Supplementary Figures (or present quantification of the different ns).

4) The authors should describe in details the different compounds used in the study the first time they are mentioned in the text (for example, PMA, MK-801, and so forth)

5) In page 6, the authors mention "By contrast, the number of double-labeled NG2+MAG+ cells remain unchanged (data not shown)." It would be unusual to observe OPCs (NG2+) with markers of terminal differentiation (MAG+ cells), I guess this is a type-O?

6) In Figure 2c, the authors mention in the figure legend that a histogram is presented, but in the figure there is a bar plot with the same data. The authors should replace it with a histogram plot.

7) The authors should discuss what might be the functional significance if the calcium oscillations observed in Figure 4, and how they might be induced.

8) In Figure 6C, I would advice the authors to show in two separate graphs the results from D-Asp I and D-Asp II experimental setups. I would also integrate table I in this figure, so it is easier to follow the experimental set-up. Also, it should be clarified which statistical methods was used throughout the figure and to which comparison the red asterisks refer to (the statistical methods used should be specified in each figure legend, and not only in the methods.

9) Does D-Asp have an effect on oligodendrocyte survival upon cuprizone treatment? Could this explain the results observed in Figure 7? The study would be benefit greatly if this would be investigated.

10) In Fig. 8E and F, the authors observe effects of D-Asp in axons with short diameter. Can the authors hypothesise why this is the case?

1st Revision - authors' response

8 October 2018

Response to Reviewer's

We thank the Reviewer's and the Editor for giving us the opportunity to revise our manuscript. We are grateful for the insightful and helpful comments provided by the Reviewer's that have significantly strengthened the manuscript.

Referee #1 (Comments on Novelty/Model System for Author):

The authors should consider using an alternative model of demyelination and remyelination, to that of the cuprizone model, to test the effects of d-aspartate on myelin damage and repair. To verify that the effect is not specific to cuprizone mediated damage.

Referee #1 (Remarks for Author):

In this study de Rosa and colleagues investigate the role of d-aspartate signaling in regulating OPC fate and subsequently the role of d-aspartate in demyelination and remyelination. The authors used an impressive array of approaches, from human to rodent in vitro cultures, organotypic cell cultures, electrophysiology, calcium imaging, behavioural and in vivo models of myelin damage and repair to identify the role of d-aspartate on OPCs fate, its mechanisms of action and its potential as a therapeutic agent for myelin disorders. Overall this is an important study in a very topical area.

The main findings were that d-aspartate treatment, in vitro, enhances OPC cell cycle time, myelin gene expression, and differentiation into myelinating oligodendrocytes. They identify d-aspartate evokes intracellular calcium rise via activation of NMDA and AMPA receptors, and subsequently via the Na/Ca exchanger. Lastly, they show that d-aspartate rescues the loss of motor function and myelin damage in the cuprizone model, a model of demyelination. As d-aspartate can be safely used in humans, this study has important therapeutic potential. However, the fact that the therapeutic effect was only observed when d-aspartate was given in the presence of cuprizone questions its mode of action.

Comment #1

- The finding that d-aspartate is not beneficial after cuprizone treatment, but only when given in the presence of the copper chelator cuprizone indicates that d-aspartate is protective against cuprizone-mediated OL death and demyelination than in improving OPC differentiation and remyelination. It is difficult to comprehend the lack of an effect on remyelination, given the potentiating effect that d-aspartate has on OPC differentiation in vitro. Therefore, to substantiate whether d-aspartate is only protective against cuprizone mediated toxicity (not unlikely given the complicated effects of copper on glutamate receptor signaling and in particular NMDA receptors) or has a generic protective role, the authors should try to use another animal model of myelin repair, such as lysolecithin LPC or equivalent model which mode of damage does not interfere with glutamate or ncx signalling. The authors should also verify whether remyelination would be evident if the animals were kept longer post-cuprizone treatment thus allowing for an investigation into the role of d-aspartate on remyelination and on the cellular mechanism of repair, and thus separating the role of d-aspartate on damage and repair?

Answer:

As properly point out by the Referee, we found that oral D-Asp treatment in mice was ineffective on remyelination when it was delivered after cuprizone withdrawal and for a period of only 2 weeks. In this regard, it should be considered that D-Asp levels are strictly regulated by its degrading enzyme DDO (D'Aniello *et al.*, 1993; Errico *et al.*, 2012), and, in line with our findings, previous studies very well demonstrated that the oral administration schedule known to substantially increase the levels of this D-aminoacid in the mouse brain is represented by chronic treatment with a 20 mM D-Asp solution for 1 month. (Errico *et al.*, 2008a and 2008b; Errico *et al.*, 2012). This observation may help to explain why, in our *in vivo* experiments, the beneficial effects of D-Asp during remyelination were observed only with long-term D-Asp treatment (3.5 weeks), and so when D-Asp treatment was initiated one week before cuprizone withdrawal. Moreover, for the same reason, it is unlikely that such beneficial effects on remyelination might be ascribed to protective actions against demyelination with only one week D-Asp treatment.

These observations have been introduced in the new version of the manuscript, discussion section, **page 18, lines 5-16**.

However, as requested by the Referee, to verify that D-Asp effect is not specific to cuprizone mediated damage, and to further investigate whether D-aspartate treatment has functional significance for remyelination we performed additional experiments using an alternative model of demyelination and remyelination such as the exposure to lysolecithin (Lysophosphatidylcholine, LPC) in cerebellar organotypic slices.

As far as concern the assessment of whether remyelination would be evident after longer time of D-Asp treatment we evaluated the effects of D-Asp treatment on remyelination at two different time points after LPC exposure. To this aim, at 11 days *in vitro* cerebellar organotypic slices were demyelinated with 0.5mg/ml LPC (Sigma) for 15-17 hours. Then, slices were washed in normal medium for 5 minutes, and treated at 2 days post lysolecithin (dpl) until 6 dpl or 10 dpl with 100 mM D-Aspartate or vehicle controls. D-Aspartate treatment of demyelinated cerebellar organotypic slices significantly upregulated MBP protein levels at 6 dpl if compared to LPC-exposed slices. D-Aspartate exposure significantly increased remyelination both at 6 dpl and 10 dpl if compared to LPC-treated slices, as measured by remyelination index (co-localization of MBP and axonal neurofilament staining, normalized to area of neurofilament). These data demonstrate that D-Aspartate exposure accelerated remyelination *in vitro*.

The results of these experiments have been introduced in the new version of Figure 2, material and methods section (**page 22, lines 10-14**), results section (**page 6, lines 18-25; page 7, lines 1-3**), discussion section (**page 18**), and corresponding legend.

Comment #2

- The effects of d-aspartate on AMPA and NMDA receptors are intriguing. However, it is not clear whether the effect of MK801 on the first intracellular calcium peak is due to the fact that MK801 is an open channel blocker and thus may not block the d-aspartate-evoked calcium until after it is applied, or whether the action of d-aspartate is via AMPA receptors, which then leads to secondary NMDA receptor effect. As result further questions arise, does d-aspartate evoke NMDA receptor mediate current in OPCs? If so would APV, a competitive agonist inhibit the calcium signal in the same way as DNQX? Then can the author explain the mode of action, if OPCs express both NMDA and AMPA receptors how come the d- aspartate-evoked current is completely blocked by AMPA antagonist?

Answer: In accordance with the Referee's comment we performed additional microfluorimetry experiments both in human oligodendrocytes MO3.13 and rat primary OPC in order to clarify the effect of D-Aspartate on NMDA receptors. To this aim Fura-2 video imaging recordings were performed following D-Asp stimulation both in human oligodendrocyte MO3.13 precursors and rat primary OPC, in presence or in absence of the competitive NMDA antagonist APV (150 μM). We found that both the selective competitive and non competitive NMDA receptor blockers, APV (150 μM) and MK-801 (10 μM), completely suppressed D-Asp induced $[Ca^{2+}]_i$ oscillations both in MO3.13 and rat primary OPCs, but only partially affected the first $[Ca^{2+}]_i$ peak (Figure 4A-B). These new results have been inserted in the new Figure 4, Results section (**page 7, lines 15-18**), and corresponding legend.

The new experiments performed, in addition to those requested by Referee's #2 in primary OPC (see also Answer to Comment #1 of Referee's #2) may help to explain D-Asp mode of action.

Our functional studies showed that D-Asp exposure elicited a complex $[Ca^{2+}]_i$ response in OPC involving an orchestrated functional crosstalk between glutamate transporters, ionotropic AMPA and NMDA glutamate receptors, and NCX3 exchangers. Indeed, while blockade of AMPA or NMDA receptors or NCX3 exchanger significantly prevented D-Asp induced $[Ca^{2+}]_i$ oscillations but only partially affected the initial $[Ca^{2+}]_i$ rise, we found that blocking glutamate transporters completely prevented both the initial and oscillatory $[Ca^{2+}]_i$ response in primary OPC. In accordance with our findings, previous studies demonstrated that the sodium-dependent glutamate transporters, beyond extracellular glutamate/D-Aspartate clearance evoked functional responses in NG2 glia (Martinez-Lozada *et al.*, 2014; Moshrefi-Ravasdjani *et al.*, 2018). In fact, intracellular sodium elevation upon activation of glutamate/D-Aspartate uptake has been associated with increased $[Ca^{2+}]_i$ signaling leading to a phosphorylation of the calcium/calmodulin-dependent kinase type II (CaMKII) and a promotion oligodendrocyte maturation (Martinez-Lozada *et al.* 2014). Moreover, our studies suggest that the action of D-Aspartate we observed on Ca^{+2} transients in primary OPC is consequent to a cooperate activation of the sodium-dependent glutamate transporter and AMPA receptors, which then leads to secondary NMDA receptor effect. Consistently, we found that D-Asp induced inward currents in primary OPC were unaffected by inhibition of NMDA receptors, but completely prevented by combined application of the glutamate transporter and AMPA inhibitors. (Discussion section, **page 14, lines 7-24**).

Minor comments:

Comment (1)

Explanation of acronym should be given in text and not only in methods, for example, PMA. Similarly, information that it activates PKC signaling, and the rationale for using PMA should also be provided in the text.

Answer: As requested, the explanation of all acronyms have been included in the text and not only in methods. In addition, we included the rationale for using PMA.

In fact, phorbol-12-myristate-13-acetate (PMA) is a natural analog of the potent protein kinase C (PKC) activator diacylglycerol and it is traditionally utilized as a tool to activate the PKC signaling pathway. Studies demonstrated that chronic PMA treatment upregulated the expression of MBP transcripts and protein in human oligodendrocyte MO3.13 cells (McLaurin *et al.*, 1995), stimulates process extension in primary oligodendrocytes (Yong *et al.* 1991), and promote their differentiation through a mechanism involving the activation of NADPH oxidase and ROS generation (Cavaliere *et al.*, 2013) (**material and methods section, page 20, lines 1-7 from the bottom**)

Comment (2)

Quantification of western blots should be provided in all figures where the blots are shown, similarly to Figure 7B.

Answer: As requested, quantification of western blots have been provided in all Figures, including protein levels of MBP in new Figure 1 and Figure 2, and GFAP and Iba1 levels in Figure 7.

Comment (3)

page 5 second line - first paragraph, ...human MO3.13 precursor or primary OPCs...' rodent or rat should be inserted in front of OPCs in this sentence.

Answer: As indicated, "primary OPC" has been changed with "rat primary OPCs.." (**page 5, line 2**)

Comment (4)

Page 4, 11th line from top, first paragraph states '...copper chelator cuprizone an in vivo model of MS...' - cuprizone model is not a model of MS. This sentence needs to be rewritten to highlight that it is a model of myelin damage and repair, an important aspect in terms of understanding the regenerative process of MS, but not a model of MS.

Answer: As correctly point out, the statement “in vivo model of MS” has been changed with “in vivo model of myelin damage and repair”. (**page 4, line 4 from the bottom**).

Referee #2 (Comments on Novelty/Model System for Author):

The authors presenting a series of experiments that conclusively suggest that D-aspartate as potential to be used for remyelination strategies in a pathological context.

Referee #2 (Remarks for Author):

In the manuscript "D-Aspartate treatment attenuates myelin damage and stimulates myelin repair", de Rosa and colleagues investigate the role of D-aspartate in oligodendrocyte lineage progression and present data that this aminoacid can have different roles in this progression and can lead to less demyelination and more remyelination in a cuprizone mouse model. The data is solid, with the authors presenting a series of experiments that conclusively suggest that D-aspartate has potential to be used for remyelination strategies in a pathological context. Therefore, I recommend publication in EMBO Molecular Medicine, with minor revisions addressing the following points:

Comment #1

1) M03.13 cell line is not purely human oligodendrocyte precursor cell line, but rather a hybrid line result of fusion of human rhabdomyosarcoma and adult human oligodendrocytes. The authors should clearly state this in the manuscript since these cells are likely to replicate some but not all aspect of human oligodendrocyte precursor cell biology. While the authors do replicate most findings also in rat primary OPCs, the findings of Figure 1c (effects of D-aspartate in cell cycle) and Figure 5 (effects of AMPA) should be replicated in primary rat OPCs.

Answer: In the new version of the manuscripts we clearly specified that M03.13 cell line is not purely human oligodendrocyte precursor cell line (see below and discussion section, **page 15, lines 1-3 from the bottom**).

As requested, we performed additional cytofluorimetry experiments to investigate the effects of D-Asp exposure on cell cycle in rat primary OPC. Furthermore, electrophysiological and microfluorimetry recordings were performed to explore the effects of D-Asp exposure on AMPA currents in rat primary OPCs.

We found that cell cycle distribution analysis by quantitative flow cytometry on rat primary OPC exposed to D-Asp showed a significant reduction in G2/M-phase cell population if compared to untreated controls. This effect was observed by 24 hours of D-Asp exposure, and persisted at 48 and 72 hours (data not shown), thus suggesting that D-Asp treatment significantly reduced proliferation of rat primary OPC. These findings propose that different mechanism of induction of oligodendrocyte differentiation can be observed with D-Asp exposure in clonal MO3.13 precursors and OPC cultures. These results have been inserted in the new Figure 1E, material and methods section (**page 23, lines 11-13**), results section (**page 5, lines 1-7 from the bottom**) discussion section (**page 16, lines 2-10**).

Next, according to Referee’s request we performed additional electrophysiological recordings to explore the effect of D-Asp exposure on rat primary OPC. Electrophysiological experiments performed on rat primary OPC revealed that 100mM D-Asp elicited an inward current that was completely prevented by 10mM DNQX in MO3.13 progenitors, but not in rat primary OPC (Figure 5C-D). Interestingly, D-Aspartate residual inward currents in OPC were unaltered by the application of the NMDA antagonist MK-801, but completely inhibited by the glutamate transporter blocker L-trans-Pyrrolidine-2,4-dicarboxylic acid (PDC) (20 mM). These results suggest that the activation of both AMPA receptors and the glutamate transporters contribute to D-Asp inward currents in OPC. These results have been inserted in the new version of Figure 5, results section (**page 9, lines 1-4 from the bottom and page 10, lines 1-12**).

Moreover, we performed additional microfluorimetry experiments to evaluate the effect of D-Asp exposure on rat primary OPC. Indeed, in rat primary OPC, the pharmacological blocking of AMPA receptors with 1.5 μ M cyanquixaline (6-cyano-7-nitroquinixaline-2,3-dione, CNQX) (data not shown) or both AMPA and NMDA receptors with 25 μ M CNQX, respectively, abolished the $[Ca^{2+}]_i$

oscillation pattern induced by D-Asp, but did not fully prevented the initial $[Ca^{2+}]_i$ rise. In line with electrophysiological findings performed in primary OPC, the glutamate transporter blocker PDC (20 mM), fully prevented the initial $[Ca^{2+}]_i$ peak as well as $[Ca^{2+}]_i$ oscillation pattern after D-Asp exposure.

The results of these experiments have been inserted in the new Figure 6, results section (**page 10, lines 1-6 from the bottom, page 11, lines 1-2**).

Altogether the results obtained might help to explain D-Asp mode of action.

“Our functional studies showed that D-Asp exposure elicited a complex $[Ca^{2+}]_i$ response in OPC involving an orchestrated functional crosstalk between glutamate transporters, ionotropic AMPA and NMDA glutamate receptors, and NCX3 exchangers. Indeed, while blockade of AMPA or NMDA receptors or NCX3 exchanger significantly prevented D-Asp induced $[Ca^{2+}]_i$ oscillations but only partially affected the initial $[Ca^{2+}]_i$ rise, we found that blocking glutamate transporters completely prevented both the initial and oscillatory $[Ca^{2+}]_i$ response in primary OPC. In accordance with our findings, previous studies demonstrated that the sodium-dependent glutamate transporters, beyond extracellular glutamate/D-Aspartate clearance evoked functional responses in NG2 glia (Martinez-Lozada *et al.*, 2014; Moshrefi-Ravasdjani *et al.*, 2019). In fact, intracellular sodium elevation upon activation of glutamate/D-Aspartate uptake has been associated with increased $[Ca^{2+}]_i$ signaling leading to a phosphorylation of the calcium/calmodulin-dependent kinase type II (CaMKII) and a promotion oligodendrocyte maturation (Martinez-Lozada *et al.* 2014). Moreover, our studies suggest that the action of D-Aspartate we observed on Ca^{+2} transients in primary OPC is consequent to a cooperate activation of the sodium-dependent glutamate transporter and AMPA receptors, which then leads to secondary NMDA receptor effect. Consistently, we found that D-Asp induced inward currents in primary OPC were unaffected by inhibition of NMDA receptors, but completely prevented by combined application of the glutamate transporter and AMPA inhibitors (**Discussion section, page 14, lines 7-24**).

In line with the key role of AMPA receptors in mediating D-Asp effects on oligodendrocyte progenitors, we found that blocking AMPA receptors completely prevented both the initial and oscillatory $[Ca^{2+}]_i$ response as well as D-Aspartate-induced inward currents in human MO3.13 oligodendrocyte progenitors. The full abrogative effect of AMPA receptor blockade on D-Asp-induced $[Ca^{2+}]_i$ in MO3.13 progenitors if compared to primary OPC might be explained by several reasons. In fact, it should be taken into consideration that M03.13 cell line is not purely human oligodendrocyte precursor cell line, but rather a hybrid line result of fusion of human rhabdomyosarcoma and adult human oligodendrocytes (Mc Laurin *et al.*, 1995) and they are likely to replicate some but not all aspect of human oligodendrocyte precursor cell biology. In addition, our results may also suggest that undifferentiated MO3.13 progenitors, at least at very early stages, may differ from OPC for the functional expression of glutamate/D-Aspartate transporters. Although this aspect required further investigation, this observation could help to explain the cycling behaviour of MO3.13 cell line compared to primary OPC after D-Asp exposure. In fact, it has been demonstrated that the absence of glutamate transporters contributes to glutamate-induced proliferative signaling (Vanhouette & Hermans, 2008). Whether the cell cycling effects of D-Asp observed in the present study maybe dependent to the functional expression of glutamate transporters need to be explored. (**discussion section, page 15, lines 1-7 from bottom, page 16, lines 1-10**).

Comment#2

2) The effects of D-aspartate in cell cycle (Fig. 1c) appear to be transient (only at day 3) and I am not convinced that they are relevant. Why is the effect only observed after 3 days? The authors observe a stabilization of cell numbers with D-Asp treatment at day 4, but what happens subsequently? Is the number of cells in control also stabilized at day 5 and beyond, or do they continue to increase while they remain stable with D-asp treatment? Addition of these time points would help clarify the role of D-Asp in proliferation. Also, according to the figure legend, n=2 in these assays, which should be increased.

Answer: In accordance with Referee’s comment we performed additional experiments to evaluate MO3.13 cell growth at 4 and 5 days, in absence or in presence of D-Asp. Analysis of cell growth revealed that the density of D-Asp-treated cells on day 3 was significantly higher compared to

untreated cells. After 4 days, the number of D-Asp-treated cells, but not those of untreated, remain unaltered compared to the number of cells recorded at 3 days. At later time points, after 5 days, the number of D-Asp-treated cells, as well as those of untreated cultures, remained stable compared to the cell number recorded at 4 days.

The results of these experiments have been introduced in the new Figure 1C and corresponding legend and in results section (**page 5, lines 10-15**).

In addition, the number of cytofluorimetry experiments in MO3.13 cells was substantially increased from N=2 to N= 5. The results obtained confirmed the already described effects of D-Asp exposure on MO3.13 cell cycle at 1, 2 and 3 days (Figure 1D). For comments related to the effects of D-Asp on cell cycle see also answer to Comment #1 and discussion section (**page 16, lines 2-10**).

Comment #3

In the western blots presented, the control lanes are separated from the D-Asp treatment. I assume these western blots are from the same gel but with intermediate bands missing, the authors should present the whole blot in Supplementary material. It is also not clear from the figure legends how many replicates the presented western blots are representative of. In addition, in Fig. 1C, the band corresponding to 100uM is narrower than the other bands. I would advice the authors to present in the main figure another western blot where all the lanes have the same width, and the remaining western blots as Supplementary Figures (or present quantification of the different ns).

Answer: As requested, we performed additional Western Blotting experiments and quantification of the different ns has been presented in each Figure. The number of replicates has been included in the corresponding legend. In addition, in Figure 1C another Western blot with lanes have similar width has been presented. Western blots in which control lanes are separated but are from the same gel (Figure 2D) were included in Supplementary material.

Comment #4

The authors should describe in details the different compounds used in the study the first time they are mentioned in the text (for example, PMA, MK-801, and so forth)

Answer: Compound used in the study have been described in detail the first time they are mentioned.

Comment #5

In page 6, the authors mention "By contrast, the number of double-labeled NG2⁺MAG⁺ cells remain unchanged (data not shown)." It would be unusual to observe OPCs (NG2⁺) with markers of terminal differentiation (MAG⁺ cells), I guess this is a type-O?

Answer: As suggested, in page 6, we replace the statement "By contrast, the number of double-labeled NG2⁺MAG⁺ cells remain unchanged (data not shown)" with "By contrast, the number of double-labeled NG2⁺MAG⁺ cells, presumably O4⁺ cells, remain unchanged (data not shown)." (**page 7, lines 3-4 from the bottom**)

Comment #6

In Figure 2c, the authors mention in the figure legend that a histogram is presented, but in the figure there is a bar plot with the same data. The authors should replace it with a histogram plot.

Answer: As requested, in Figure 2c the bar plot has been replaced with an histogram plot.

Comment #7

The authors should discuss what might be the functional significance of the calcium oscillations observed in Figure 4, and how they might be induced.

Answer: Indeed, agonist-evoked $[Ca^{2+}]_i$ oscillations are a characteristic property of cells expressing some receptors, including AMPA receptors, and represent a signaling system that regulates numerous processes in all cell types including proliferation and cellular differentiation (Dolmetsch *et al.*, 1998). Recently, Krasnow *et al.*, (2017) provide evidence that calcium transients in developing oligodendrocytes, including those evoked by neuronal activity, drive myelin sheath

elongation presumably controlling proteins regulating cytoskeletal growth and myelin assembly. In this context, and in line with our findings showing the relevant contribution of NCX3 to D-Asp-evoked $[Ca^{2+}]_i$ oscillations in OPC, a very recent study demonstrated that NCX-mediated Ca^{+2} influx is required for sustaining spontaneous $[Ca^{2+}]_i$ oscillations occurring in differentiating oligodendrocytes at DIV4-5 in cultures (Hamman *et al.*, 2018). Based on these observations, we can speculate that D-Asp exposure, by promoting $[Ca^{2+}]_i$ oscillations may shift the onset of spontaneous calcium activity to earlier time period, thus triggering the developmental programme and accelerating oligodendrocyte differentiation.

These observations have been inserted in the discussion section of the new version of the manuscript (page 17, lines 1-12).

As far as concern how Ca^{+2} oscillations might be induced, see also Answer to Comment #2 of Referee #1: "...our studies suggest that the action of D-Aspartate we observed on Ca^{+2} transients in primary OPC is consequent to a cooperate activation of the sodium-dependent glutamate transporter and AMPA receptors, which then leads to secondary NMDA receptor effect. Consistently, we found that D-Asp induced inward currents in primary OPC were unaffected by inhibition of NMDA receptors, but completely prevented by combined application of the glutamate transporter and AMPA inhibitors" (page 14, lines 2-7 from the bottom).

Comment #8

In Figure 6C, I would advice the authors to show in two separate graphs the results from D-Asp I and D-Asp II experimental setups. I would also integrate table I in this figure, so it is easier to follow the experimental set-up. Also, it should be clarified which statistical methods was used throughout the figure and to which comparison the red asterisks refer to (the statistical methods used should be specified in each figure legend, and not only in the methods).

Answer: As suggested by the Referee the results from D-Asp I and D-Asp II experimental setups are now shown in two separate graphs in Figure 6C. In addition, table I has been integrated in this figure. Furthermore, in the legend of Figure 6C we specified that red asterisks indicate significance of D-Asp treated animals versus cuprizone-treated mice during demyelination or vehicle-treated mice during remyelination. The statistical methods used were specified in the corresponding legend of all figures.

Comment #9

Does D-Asp have an effect on oligodendrocyte survival upon cuprizone treatment? Could this explain the results observed in Figure 7? The study would be benefit greatly if this would be investigated.

Answer: To investigate the effects of D-Asp on oligodendrocyte survival upon cuprizone treatment and on oligodendrocyte maturation during *corpus callosum* remyelination, we analyzed the number of cells immunostained for Olig2, a transcription factor expressed in all cell types of the oligodendrocyte lineage, and the number of Olig2⁺ cells co-expressing adenomatous polyposis coli CC1, a marker of mature oligodendrocytes. Quantitative colocalization experiments performed in the *corpus callosum* of cuprizone-treated mice, in absence or in presence of D-Asp for 5 weeks, showed that the number of both Olig2⁺ cells and those coexpressing CC1 (Olig2⁺/CC1⁺ cells) were significantly higher in D-Asp-treated mice compared to cuprizone-fed animals. This suggests that D-Asp treatment during demyelination has a protective role on oligodendrocytes, thus explaining its attenuating effects on myelin damage.

These new results have been introduced in the new Figure 9A, corresponding legend and Results section (page 12, last three lines, and page 13 lines 1-7).

Comment #10

In Fig. 8E and F, the authors observe effects of D-Asp in axons with short diameter. Can the authors hypothesise why this is the case?

Answer: Remarkable, D-Asp treatment significantly increased the percentage of myelinated axons with small diameter (0.2-0.4mm) during remyelination. Recent findings showed that an efficient remyelination of smaller-diameter axons depends on neuronal activity more than larger diameter

axons and on the activation of glutamate ionotropic receptors between demyelinated axons and OPCs (Gautier *et al.*, 2015). In line with this observation and beside the direct effects of D-Asp on OPC we observed in the present study, we can speculate that the stimulatory effect of D-Asp on myelination of small-diameter axons may be dependent to its ability to stimulate neuronal activity. In fact, previous studies very well demonstrated that D-Asp treatment increased neuronal-activity-dependent synaptic plasticity and glutamate release (Errico *et al.*, 2008b; Sacchi *et al.*, 2017). These observations have been introduced in the Discussion section of the new version of the manuscript (page 18, last nine lines).

2nd Editorial Decision

6 November 2018

Thank you for the submission of your revised manuscript to EMBO Molecular Medicine. We have now received the enclosed reports from the referees that were asked to re-assess it. As you will see the reviewers are now globally supportive and I am pleased to inform you that we will be able to accept your manuscript pending minor editorial amendments [not listed].

***** Reviewer's comments *****

Referee #1 (Remarks for Author):

The manuscript has greatly improved and the authors have addressed all my concerns.

Referee #2 (Remarks for Author):

The authors have answered my questions appropriately, so I recommend the publication of the manuscript. However, there is what appears to me a mistake in Figures 2B and 2F, the panels with higher magnification do not seem to correspond to the white squares displayed in a and e, when examining the pattern of the stainings. It might be the angle of the square, but the authors should double-check this point.

2nd Revision - authors' response

20 November 2018

Authors made the requested editorial changes.

Corresponding Author Name: Francesca Boscia

Journal Submitted to: EMBO Mol Med

Manuscript Number: EMM-2018-09278